# System-Aware Neural ODE Processes for Few-Shot Bayesian Optimization

**Jixiang Qing**[1]*, **Becky D Langdon**[1], **Robert M Lee**[2], **Behrang Shafei**[2],
**Mark van der Wilk**[3], **Calvin Tsay**[1], **Ruth Misener**[1]
*[1]Imperial College London    [2]BASF SE    [3]University of Oxford*

**Reviewed on OpenReview:** *https://openreview.net/forum?id=FFnRLvWefK*

## Abstract

We consider the problem of optimizing initial conditions and termination time in dynamical systems governed by unknown ordinary differential equations (ODEs), where evaluating different initial conditions is costly and the state's value can not be measured in real-time but only with a delay while the measuring device processes the sample. To identify the optimal conditions in limited trials, we introduce a few-shot Bayesian Optimization (BO) framework based on the system's prior information. At the core of our approach is the System-Aware Neural ODE Processes (SANODEP), an extension of Neural ODE Processes (NODEP) designed to meta-learn ODE systems from multiple trajectories using a novel context embedding block. We further develop a two-stage BO framework to effectively incorporate search space constraints, enabling efficient optimization of both initial conditions and observation timings. We conduct extensive experiments showcasing SANODEP's potential for few-shot BO within dynamical systems. We also explore SANODEP's adaptability to varying levels of prior information, highlighting the trade-off between prior flexibility and model fitting accuracy.

## 1 Introduction

This paper studies a widely encountered, yet less investigated, problem: optimizing the initial conditions and termination time in unknown dynamical systems where evaluations are computationally expensive. We assume that the primary evaluation cost comes from switching initial conditions and wish to use as few trajectories as possible.

This issue is prevalent in multiple fields, including biotechnology, epidemiology, ecology, and chemistry. For instance, consider the optimization of a (bio)chemical reactor. Here, the objective is to determine the optimal "recipe," i.e., the set of initial reactant concentrations and reaction times that maximize yield, enhance selectivity, and/or minimize waste (Taylor et al., 2023; Schoepfer et al., 2024; Schilter et al., 2024; Thebelt et al., 2022). Specifically, the high costs associated with changing reactants for multiple experimental runs highlight the need for developing efficient optimization algorithms (Paulson & Tsay, 2024).

Bayesian Optimization (BO) (Frazier, 2018; Garnett, 2023) is a well-established method for optimizing expensive-to-evaluate black-box objective functions. It relies on probabilistic surrogate models built from limited function evaluations to guide the optimization efficiently. However, standard Gaussian Processes (GPs) (Williams & Rasmussen, 2006) (the *de facto* probabilistic surrogate model in BO) with conventional kernels do not capture the dynamics of these systems effectively. Recent efforts to introduce Bayesian ODE models aim to incorporate suitable assumptions; however, their inference often involves time-consuming computations (Heinonen et al., 2018; Dandekar et al., 2020), making them impractical for a time-sensitive optimization scenario. Some Bayesian ODE models attempt to alleviate computational burdens by leveraging crude approximation inference (Ott et al., 2023) or approximating numerical integrators (Hegde et al., 2022;

---

*Corresponding author: `j.qing@imperial.ac.uk`

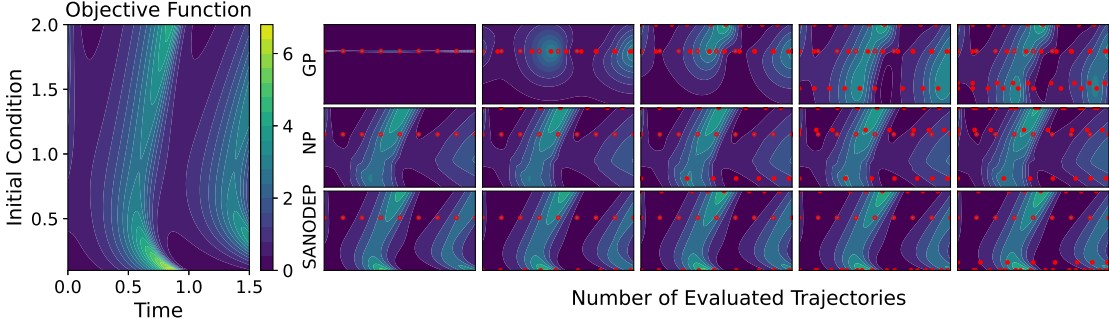

Figure 1: Illustration of meta-learning-based few-shot Bayesian Optimization (BO) with time-delay constraints (detailed in Section 4) applied to the Lotka-Volterra (LV) system using three different models: Gaussian Process (GP), Neural Process (NP), and System Aware Neural ODE Process (SANODEP) (Section. 3). Starting with one randomly sampled trajectory (marked with a ○), The BO progressively selects promising initial conditions for experiments (Section. 4.1) and, based on these, recommends subsequent measurement times while enforcing a minimum time delay constraint ($\Delta t$) (Section. 4.2). The figure demonstrates that after evaluating just one trajectory (the first, left-most column of the three rows), the meta-learned SANODEP model more closely approximates the original LV system compared to the non-meta-learned GP model. This enhanced approximation facilitates the BO search in identifying more promising initial conditions.

Ensinger et al., 2024). While these approaches reduce computation time, the necessary approximations may degrade model performance. Consequently, the widely encountered yet under-investigated problem of performing BO in dynamical systems remains open, primarily due to the lack of an appropriate model.

This work attempts to tackle this optimization problem by conceptualizing it as a *few-shot* optimization, through leveraging *prior information* about dynamical systems to formulate the *meta tasks* to train a learning model, which is then able to adapt to new problems with very few data (a.k.a., shots) and served as our BO surrogate model. To achieve this, we focus on the Neural ODE Process (NODEP) (Norcliffe et al., 2021) as our learning model due to several useful model properties. First, NODEP combines Neural ODEs (Chen et al., 2018)—emphasizing the dynamical system perspective that may provide a more informative inductive bias—with Neural Processes (NP) (Garnelo et al., 2018b), enhancing the meta-learning aspect for few-shot optimization. Second, meta-learning endows NODEP with fast adaptation to new observations, mitigating the potential concern of model training time with incoming measurement results. Third, NODEP enables continuous gradient-based optimization over time, compared with discrete-time meta-learn models (e.g., Foong et al. (2020)), integrating seamlessly within the BO framework. Finally, compared to other continuous-time models involving computationally intensive operations (e.g., time domain attention (Chen et al., 2024)), NODEP is computationally efficient, making it suitable for BO.

However, building a meta-learning-based few-shot BO framework for unknown dynamical systems is not a straightforward downstream application. An obstacle is that NODEP's functionality is limited by its original learning objective: predicting a single *trajectory* state value given *context* data for that trajectory. For initial condition optimization of unknown ODE systems, it is essential to learn to be aware of the underlying governing systems using context data consisting of one or several different trajectory observations, as this benefits for forecasting trajectories starting from arbitrary initial conditions. Moreover, no BO frameworks are proposed for such problem settings.

In line with our optimization requirements, we introduce the System-Aware Neural ODE Process (SANODEP), a generalization of NODEP in terms of meta-learning of ODE systems to plug-in our newly developed BO framework for few-shot BO in dynamical systems. Figure 1 shows, through example, SANODEP outcomes in comparison to other approaches. Our contributions are as follows:

1. We propose a novel context embedding mechanism enabling SANODEP to meta-learn from batch of trajectories with minimal model structure adjustments based on NODEP.

2. We developed a model-agnostic, time-delay constraint process BO framework for optimizing initial conditions and termination time in dynamical systems.

3. We compare SANODEP and other meta-learned models under our BO framework, validating the benefits of the ODE-aware model structure and demonstrating the effectiveness of the few-shot BO framework's model agnosticity property.

4. We conduct initial investigation on how the different levels of prior information can be utilized for SANODEP. Strong prior information can enable a physically-informed model structure with extended loss considering parameter inference, enabling a novel *few-shot parameter estimation* functionality, while weak prior information may still be useful through a properly designed *task distribution*, albeit at a detriment to model fitting capability.

The rest of the paper is organized as follows: In Section 2, we describe the preliminaries. Section 3 introduces the System-Aware Neural ODE Processes (SANODEP). In Section 4, we develop the optimization framework based on SANODEP specific to our optimization problem. Section 5 discusses related work. Section 6 presents the numerical experiments we conducted. In Section E, we investigate the impact of different levels of prior information.

## 2 Problem Statement and Background

### 2.1 State Optimization

Consider a dynamical system whose evolution is described by the following ordinary differential equation (ODE) system in the time domain, denoted as $t$:

$$\frac{d\boldsymbol{x}}{dh} = \boldsymbol{f}(\boldsymbol{x}, h), \quad \boldsymbol{x}(h = t_0) = \boldsymbol{x}_0, \tag{1}$$

where $\boldsymbol{x}(t) \in \mathcal{X}_{\boldsymbol{x}} \subset \mathbb{R}^{d_{\boldsymbol{x}}}$ represents the state value of the system at time $t$, $\boldsymbol{f}(\cdot) : \mathbb{R}^{d_{\boldsymbol{x}}} \times \mathbb{R} \to \mathbb{R}^{d_{\boldsymbol{x}}}$ is an unknown function (*vector field*) governing the system dynamics, and $\boldsymbol{x}_0 \in \mathcal{X}_0 \subseteq \mathcal{X}_{\boldsymbol{x}} \subset \mathbb{R}^{d_{\boldsymbol{x}}}$ is the initial system state. Define $\boldsymbol{f}_{evolve}(\boldsymbol{x}_0, t, \boldsymbol{f}) : \mathcal{X}_0 \times \tau \times (\mathbb{R}^{d_{\boldsymbol{x}}} \times \mathbb{R} \to \mathbb{R}^{d_{\boldsymbol{x}}}) \to \mathbb{R}^{d_{\boldsymbol{x}}}$, where $\tau \in [t_0, t_{max}]$, to obtain the system state at time $t$ from an initial condition $\boldsymbol{x}_0$ as: $\boldsymbol{f}_{evolve}(\boldsymbol{x}_0, t, \boldsymbol{f}) = \boldsymbol{x}_0 + \int_{t_0}^{t} \boldsymbol{f}(\boldsymbol{x}(h), h)dh$. Let $g(\cdot) : \mathbb{R}^{d_{\boldsymbol{x}}} \to \mathbb{R}$ be a practitioner-specified (known) function that aggregates the state values into a scalar, we consider the following multi-objective problem:

$$\max_{\{t, \boldsymbol{x}_0\} \in \tau \times \mathcal{X}_0} \quad g\left(\boldsymbol{f}_{evolve}(\boldsymbol{x}_0, t, \boldsymbol{f})\right), -t. \tag{2}$$

The goal is to identify the initial conditions $\boldsymbol{x}_0$ along with the corresponding evolution termination time $t$ that provide the optimal trade-off between the objective $g$ applied to the state values $\boldsymbol{x}(t)$ and the amount of time to reach this state. Simply put, we wish to maximize the objective function $g$ early.

### 2.2 Neural ODE Processes (NODEP)

NODEP (Norcliffe et al., 2021) is a latent variable-based Bayesian meta-learning model with generative processes that can be summarized as follows: given a set of $N$ observations of state values at different times from a single trajectory, represented by context set $\mathbb{C} := \{(t_i^{\mathbb{C}}, \boldsymbol{x}_i^{\mathbb{C}})\}_{i=1}^N$, NODEP assumes the conditional prediction has been generated from a latent controlled ODE of state dimensionality $d_{\boldsymbol{\ell}}$. The stochasticity of the model is induced by stochastic latent initial condition $L_0$ and the stochastic dynamics representation, which in practice is implemented via a time-invariant control $L_D$. These two terms are obtained through first encoding the context elements $\phi_r([t_i^{\mathbb{C}}, \boldsymbol{x}_i^{\mathbb{C}}])$ and then applying average pooling (Zaheer et al., 2017) to produce a single representation vector $\boldsymbol{r} = \frac{1}{N}\sum_{i=1}^N \phi_r([t_i^{\mathbb{C}}, \boldsymbol{x}_i^{\mathbb{C}}])$, which is then mapped to corresponding distributions $L_0$ and $L_D$. Once we sample the latent initial condition and the trajectory dynamics representation $\boldsymbol{u}^1$, the

---

[1]With slight abuse of notation, we follow the common notation in control to use $\boldsymbol{u}$ (instead of $\boldsymbol{d}$) to represent the realization of time-invariant control term $L_D$.

evolution of the latent ODE is:

$$\boldsymbol{l}(t) = \boldsymbol{f}_{evolve}(\boldsymbol{l}_0, t, \boldsymbol{f}_{nn}(\boldsymbol{u}, \theta_{ode})) := \boldsymbol{l}(t_0) + \int_{t_0}^{t} \boldsymbol{f}_{nn}\left(\boldsymbol{l}(h), \boldsymbol{u}, h, \theta_{ode}\right) dh. \tag{3}$$

Here $\boldsymbol{f}_{nn}$ represents the vector field as a neural network parameterized by $\theta_{ode}$. We compute $\boldsymbol{f}_{evolve}$ using numerical ODE solvers. To predict the state values at time $t$, after solving the latent ODE, the latent state $\boldsymbol{l}(t) \in \mathbb{R}^{d_l}$ can be decoded back to real state predictions through a decoder $p(\boldsymbol{x}(t)|\boldsymbol{l}(t), t)$. Thus, the overall joint probability of the model can be represented as:

$$p\left(\boldsymbol{u}, \boldsymbol{l}_0, \boldsymbol{x}|\mathbb{C}, t\right) = p\left(\boldsymbol{x}|\boldsymbol{l}_0, \boldsymbol{u}, t\right) p(\boldsymbol{u}|\mathbb{C})p(\boldsymbol{l}_0|\mathbb{C}). \tag{4}$$

The latent state's dimensionality $d_l$ is typically chosen to be larger than $d_{\boldsymbol{x}}$ as additional state variables benefit the model flexibility (Dupont et al., 2019) and enable learning of higher-order dynamics implicitly (Norcliffe et al., 2020).

Given a context set $\mathbb{C}$ and a target set $\mathbb{T} := \{(t_k^{\mathbb{T}}, \boldsymbol{x}_k^{\mathbb{T}})\}_{k=1}^{J}$ [2], to calculate the log-likelihood on $\mathbb{T}$ for inference, the intractable posterior of $p\left(\boldsymbol{l}_0|\mathbb{C}\right) p\left(\boldsymbol{u}|\mathbb{C}\right)$ has been approximated through the mean field approximation $q(\boldsymbol{\ell}_0|\mathbb{T})q(\boldsymbol{u}|\mathbb{T})$, which in practice has been implemented through encoder $q_L\left(\boldsymbol{l}_0|\mathbb{C}\right)$, $q_D\left(\boldsymbol{u}|\mathbb{C}\right)$ in an amortized fashion, eventually leading to the following evidence lower bound (ELBO):

$$\begin{aligned}
&\log p\left(\{(x_k^{\mathbb{T}})\}_{k=1}^{J}|\{(t_i^{\mathbb{C}}, \boldsymbol{x}_i^{\mathbb{C}})\}_{i=1}^{N}, \boldsymbol{T}^{\mathbb{T}}\right) \\
&\approx \mathbb{E}_{q(\boldsymbol{\ell}_0, \boldsymbol{u}|\mathbb{T})}\left[\sum_{k=1}^{J} \log p\left(\boldsymbol{x}_k|\boldsymbol{u}, \boldsymbol{l}_0, t_k\right)\right] - \mathrm{KL}\left[q_L\left(\boldsymbol{l}_0|\mathbb{T}\right)||q_L\left(\boldsymbol{l}_0|\mathbb{C}\right)\right] - \mathrm{KL}\left[q_D\left(\boldsymbol{u}|\mathbb{T}\right)||q_D\left(\boldsymbol{u}|\mathbb{C}\right)\right],
\end{aligned} \tag{5}$$

where $\boldsymbol{T}^{\mathbb{T}} := \{(t_k^{\mathbb{T}})\}_{k=1}^{J}$ represents the (irregularly sampled) target times[3]. NODEP has competitive performance in predicting single trajectory, and we therefore consider extending its functionality to meta-learn ODE system distributions for our subsequent optimization purpose.

## 3 System Aware Neural ODE Process

**Few-Shot Optimization with Prior Information**
Given that evaluating $\boldsymbol{f}_{evolve}(\boldsymbol{x}_0, t, \boldsymbol{f})$ is limited by cost to a small number of different initial conditions. To enable fast adaptation with few evaluations, we take a meta-learning approach by assuming that $\boldsymbol{f}$ is a realization of a stochastic function $\mathcal{F}$, and we can access data from its different realizations to formulate the *meta-task distributions*.

More precisely, consider a random function $\mathcal{F} : \mathcal{X}_0 \times \tau \to \mathbb{R}^{d_{\boldsymbol{x}}}$ representing the distribution of dynamical systems. From a specific realization of $\mathcal{F}$, suppose we have observed $M$ distinct trajectories, the context set then encompasses observations from all $M$ trajectories, denoted as $\mathcal{T}^{\mathbb{C}} = \{\mathcal{T}_1^{\mathbb{C}}, ..., \mathcal{T}_M^{\mathbb{C}}\}$. Each trajectory, labeled as $\mathcal{T}_l^{\mathbb{C}}$, includes its own set of $N_l$ context observations: $\mathcal{T}_l^{\mathbb{C}} = \{(t_{l_i}^{\mathbb{C}}, \boldsymbol{x}_{l_i}^{\mathbb{C}})\}_{i=1}^{N_l}$. Furthermore, consider a *new* trajectory $\mathcal{T}_{new}$. After observing additional context observations on this trajectory, $\mathcal{T}_{new}^{\mathbb{C}}$ (e.g., the initial condition of this new trajectory), we can define the extended context set as $\mathbb{C} = \mathcal{T}^{\mathbb{C}} \cup \mathcal{T}_{new}^{\mathbb{C}}$.

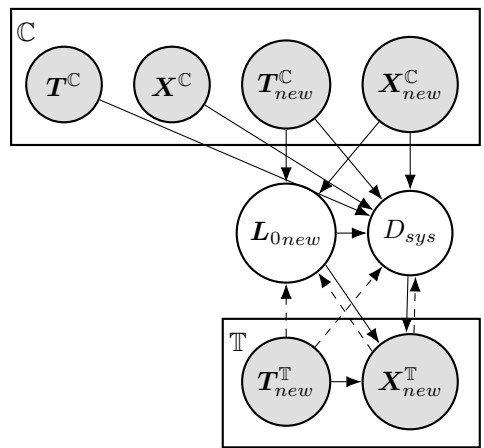

Figure 2: Graphical Model of SANODEP, the model predicts any time point in the new trajectory by knowing both observations from new trajectories and from past trajectories. Depending on whether the $X_{new}^{\mathbb{C}}$ and $T_{new}^{\mathbb{C}}$ consists of more than the initial condition, the model focuses on forecasting or interpolating tasks. The solid and dashed lines represent the generative and inference processes, respectively.

---

[2] We follow the Garnelo et al. (2018b) and Norcliffe et al. (2021) convention by assuming $\mathbb{T}$ is a superset of $\mathbb{C}$.

[3] We denote context trajectory data either as a set of pairs $\{(t_i^{\mathbb{C}}, \boldsymbol{x}_i^{\mathbb{C}})\}_{i=1}^{N}$ or as vectors $\{\boldsymbol{T}^{\mathbb{C}}, \boldsymbol{X}^{\mathbb{C}}\}$, where $\boldsymbol{T}^{\mathbb{C}} = [t_1^{\mathbb{C}}, ..., t_N^{\mathbb{C}}]^{\mathrm{T}}$, $\boldsymbol{X}^{\mathbb{C}} = [\boldsymbol{x}_1^{\mathbb{C}}, ..., \boldsymbol{x}_N^{\mathbb{C}}]^{\mathrm{T}}$. We use similar notation $\{\boldsymbol{T}^{\mathbb{T}}, \boldsymbol{X}^{\mathbb{T}}\}$ for the target set.

Additionally, the target set is defined as $\mathbb{T} = \mathcal{T}^{\mathbb{C}} \cup \mathcal{T}_{new}^{\mathbb{T}}$[4], a visual elaboration is provided in Figure 5 in Appendix A.

It is straightforward to see why NODEP is suboptimal in such scenario as it makes predictions based on single time series ($\mathbb{C} = \mathcal{T}_{new}^{\mathbb{C}}$, $\mathbb{T} = \mathcal{T}_{new}^{\mathbb{T}}$), unable to leverage $M$ trajectorys' information. Below, we demonstrate how SANODEP is efficiently enabled through the set-based representation (Zaheer et al., 2017).

### 3.1 Set-based Dynamical System Representation

Similar to the latent variable $\boldsymbol{u}$ (Eq. 3) capturing trajectory dynamics in NODEP, we adapt a latent variable $D_{sys} \sim q(\boldsymbol{u}_{sys}|\mathbb{C})$ in SANODEP, effectively replacing $\boldsymbol{u}$ in the model structure but with an enhanced conditioning on context observations from $M + 1$ trajectories, to capture the dynamical systems properties. Follow this mechanism, we will have a feature extraction for **multi-start multivariate irregular time series**, in addition with an efficiency requirement for our optimization purposes.

Such questions are less considered in contemporary time series models, as existing approaches (Shukla & Marlin, 2021; Schirmer et al., 2022) are mainly developed for single initial condition start trajectories. Specifically for multi-start scenario, Jiang et al. (2023) proposes first extracting a trajectory-wise aggregated feature vector, and then averaging feature vectors as a final representation. However, both extraction are implemented through a convolution operation, which poses challenges with irregular time series without additional modifications.

A straightforward thinking would be still using average pooling across all context elements as a set-based representation. While the Picard-Lindelöf theorem (Lindelöf, 1894) guarantees the uniqueness of state values in an initial value problem when $\boldsymbol{f}(\boldsymbol{x}, t)$ is Lipschitz continuous, in case when only part of the underline system states can be measured as $\boldsymbol{x}(t)$, different trajectories might still share identical $\boldsymbol{x}(t)$ values, leading to identifiability issues (duplicate context elements come from different initial conditions). Consequently, we propose an augmented sets-based approach: we augment each observation with its corresponding initial condition: $(t_i, \boldsymbol{x}_0, \boldsymbol{x}_i)$, to enhance the model's ability to differentiate between trajectories that might otherwise appear identical. Then we perform the average pooling on the flattened context set to obtain the context representation $\boldsymbol{r}_{sys} = \frac{1}{\sum_{l=1}^{M+1} N_l} \sum_{l=1}^{M+1} \sum_{i=1}^{N_l} \phi_{r_{sys}}([t_{l_i}^{\mathbb{C}}, \boldsymbol{x}_{l_0}^{\mathbb{C}}, \boldsymbol{x}_{l_i}^{\mathbb{C}}])$, which is then mapped to a distribution representing possible system realizations from $\mathcal{F}$ that has generated $\mathbb{C}$. Aside from the theoretical intuition, we also empirically compare the average pooling without initial condition augmentation in the Section. 6.1, and find that the augmentation leads to more robust performance even $\mathcal{F}$ only represents first-order systems with fully observable states.

For the rest of the model, SANODEP follows the NODEP structure, hence requiring minimal adjustment, only with additional care on activation function choices to enable differentiability (Appendix A.3). Appendix A.2 provides a detailed description of the model structure.

### 3.2 Bi-scenario Loss Function

In line with the principles of episode learning (Vinyals et al., 2016), SANODEP's training is structured through multiple episodes. Motivated by our optimization problem that will be detailed in Section. 4, the design of each episode's problem ensures good model performance under two primary scenarios:

1. **Forecasting**: Using $M$ context trajectories $\mathcal{T}^{\mathbb{C}}$, the model predicts future state values $\boldsymbol{X}_{new}^{\mathbb{T}}$ for a new trajectory initiated from $\mathcal{T}_{new}^{\mathbb{C}} = \{(t_{new_0}^{\mathbb{C}}, \boldsymbol{x}_{new_0}^{\mathbb{C}})\}$ at designated target times $\boldsymbol{T}_{new}^{\mathbb{T}}$.

2. **Interpolating**: From the same $M$ trajectories, the model interpolates and extrapolates state values for a new trajectory that already includes $K > 1$ observations: $\mathcal{T}_{new}^{\mathbb{C}} = \{(t_{new_i}^{\mathbb{C}}, \boldsymbol{x}_{new_i}^{\mathbb{C}})\}_{i=0}^{K}$, predicting the states at times $\boldsymbol{T}_{new}^{\mathbb{T}}$.

---

[4]Note that again $\mathcal{T}_{new}^{\mathbb{C}} \subset \mathcal{T}_{new}^{\mathbb{T}}$.

Both scenarios can be considered under a bi-scenario loss function, designed to enhance model's accuracy for predicting new trajectory states $\boldsymbol{X}_{new}^{\mathbb{T}}$ at $\boldsymbol{T}_{new}^{\mathbb{T}}$:

$$\mathcal{L}_\theta = \mathbb{E}_{\boldsymbol{f} \sim \mathcal{F}, M, \mathcal{T}^{\mathbb{C}}, \mathcal{T}_{new}^{\mathbb{T}}, \mathbb{1}_{forecast}, \mathcal{T}_{new}^{\mathbb{C}}(\mathbb{1}_{forecast})} \log p_\theta \left( \boldsymbol{X}_{new}^{\mathbb{T}} | \mathcal{T}^{\mathbb{C}} \cup \mathcal{T}_{new}^{\mathbb{C}}(\mathbb{1}_{forecast}), \boldsymbol{T}_{new}^{\mathbb{T}} \right), \quad (6)$$

where $p_\theta(\cdot)$ represents the SANODEP prediction when parameterized by $\theta$.

---

**Algorithm 1** Learning and Inference in System Aware Neural ODE Processes (SANODEP)

---

**Require:** ODE system inducing distributions $P$, known trajectory range $[M_{min}, M_{max}]$, batch size of Monte Carlo approximation of initial condition sample $N_{\boldsymbol{x}_0}$, batch size of dynamical systems sample $N_{sys}$, prespecified time grid $\boldsymbol{T}_{grid} := \texttt{linspace}(t_0, t_{max}, N_{grid})$, minimum and maximum context points within a trajectory $m_{min}, m_{max}$. minimum and maximum extra target points $n_{min}, n_{max}$, initial condition space $\mathcal{X}_0$.

1:   Initialize SANODEP model parameters $\theta$ with random seeds.
2:   **for** step in `training_steps` **do**
3:     # Training data generation
4:     **for** $j = 1$ to $N_{sys}$ **do**
5:       Sample ODE system $\boldsymbol{f} \sim \mathcal{F}$ and $N_{\boldsymbol{x}_0}$ different initial conditions from $\mathcal{X}_0$, solve $N_{\boldsymbol{x}_0}$ odes at time grids $\boldsymbol{T}_{grid}$ to obtain the dataset $\{\mathcal{T}_1, ..., \mathcal{T}_{N_{\boldsymbol{x}_0}}\}$.
6:       Sample known trajectory number $M$ from $\texttt{Uniform}(M_{min}, M_{max})$.
7:       **for** $l = 1$ to $M$ **do**
8:         # Sample context and target elements within each trajectory
9:         Sample $m_l$ from $\texttt{Uniform}(m_{min}, m_{max})$, sample $n_l$ from $\texttt{Uniform}(n_{min}, n_{max})$.
10:        Randomly subsample from $\mathcal{T}_l$ to extract the context dataset $\mathcal{T}_l^{\mathbb{C}}$ and target dataset $\mathcal{T}_l^{\mathbb{T}}$, where $|\mathcal{T}_l^{\mathbb{C}}| = m_l$ and $|\mathcal{T}_l^{\mathbb{T}}| = m_l + n_l$ and $\mathcal{T}_l^{\mathbb{C}} \subseteq \mathcal{T}_l^{\mathbb{T}}$.
11:       **end for**
12:       Concatenate $M$ trajectories context set $\mathcal{T}^{\mathbb{C}} = \{\mathcal{T}_1^{\mathbb{C}}, ..., \mathcal{T}_M^{\mathbb{C}}\}$ and target set $\mathcal{T}^{\mathbb{T}} = \{\mathcal{T}_1^{\mathbb{T}}, ..., \mathcal{T}_M^{\mathbb{T}}\}$.
13:     **end for**
14:     # Model Prediction
15:     **for** $j = 1$ to $N_{sys}$ **do**
16:       **for** $k = 1$ to $N_{\boldsymbol{x}_0}$ **do**
17:         sample $\mathbb{1}_{forecast}$ from $\texttt{Bernoulli}(\lambda)$  # determine forecasting or interpolating
18:         **if** $\mathbb{1}_{forecast} = 1$ **then**
19:           $\mathcal{T}_{new}^{\mathbb{C}} = \{(t_{0k}, \boldsymbol{x}_{0k})\}$ where $(t_{0k}, \boldsymbol{x}_{0k}) \in \mathcal{T}_k$
20:         **else if** $\mathbb{1}_{forecast} = 0$ and $k > M$ **then**
21:           Randomly subsample from $\mathcal{T}_k$ to extract the context dataset $\mathcal{T}_k^{\mathbb{C}}$ and target dataset $\mathcal{T}_k^{\mathbb{T}}$, where $|\mathcal{T}_k^{\mathbb{C}}| = m$, $|\mathcal{T}_k^{\mathbb{T}}| = m + n$ and $\mathcal{T}_k^{\mathbb{C}} \subseteq \mathcal{T}_k^{\mathbb{T}}$.
22:         **else**
23:           $\mathcal{T}_{new}^{\mathbb{C}} = \mathcal{T}_k^{\mathbb{C}}, \mathcal{T}^{\mathbb{C}} = \mathcal{T}^{\mathbb{C}} \backslash \mathcal{T}_k^{\mathbb{C}}$.
24:         **end if**
25:         Augmented the context trajectories and target trajectories with the new trajectory $\mathcal{T}^{\mathbb{C}} = \mathcal{T}^{\mathbb{C}} \cup \mathcal{T}_{new}^{\mathbb{C}}$, $\mathcal{T}^{\mathbb{T}} = \mathcal{T}^{\mathbb{T}} \cup \mathcal{T}_{new}^{\mathbb{T}}$.
26:         Compute variational posterior $q(\boldsymbol{u}_{sys}|\mathcal{T}^{\mathbb{T}})$, $q(\boldsymbol{L}_{0new}^{\mathbb{T}}|\mathcal{T}^{\mathbb{T}})$, $q(\boldsymbol{u}_{sys}|\mathcal{T}^{\mathbb{C}})$, $q(\boldsymbol{L}_{0new}^{\mathbb{T}}|\mathcal{T}^{\mathbb{C}})$ through the encoder block.
27:         Sample $\boldsymbol{l}(t_0), \boldsymbol{u}_{sys}$ from $q(\boldsymbol{l}(t_0)|\mathcal{T}^{\mathbb{T}}), q(\boldsymbol{u}_{sys}|\mathcal{T}^{\mathbb{T}})$.
28:         Solve latent odes as in Eq. (3) for all times and decode to obtain the model prediction.
29:         Calculate the trajectory wise loss $\mathcal{L}_{\text{ELBO}k}$ based on Eq. (8).
30:       **end for**
31:     **end for**
32:     Average the trajectory wise loss $\mathcal{L}_{\text{ELBO}} = \frac{1}{N_{\boldsymbol{x}_0}} \sum_{o=1}^{N_{\boldsymbol{x}_0}} \mathcal{L}_{ELBOo}$
33:     Update the model parameter through optimizer $\theta \leftarrow \theta - \eta \nabla_\theta \mathcal{L}_{\text{ELBO}}$
34:   **end for**

---

During training, once a dynamical system realization $\boldsymbol{f}$ from $\mathcal{F}$ has been drawn, we randomly sample the number of context trajectories $M$ that we have observed, where the number of context elements, the initial

condition of the trajectory and the observation time are all sampled. For the new trajectory to be predicted, besides sampling the target set $\{\boldsymbol{T}_{new}^{\mathbb{T}}, \boldsymbol{X}_{new}^{\mathbb{T}}\}$, a Bernoulli indicator $\mathbb{1}_{forecast} \sim \text{Bernoulli}(\lambda)$ parameterize by $\lambda$ is sampled to determine whether the episode will address a forecasting or interpolating scenario. This indicator directly influences the part of the context set sampled from the new trajectory[5]:

$$
\mathcal{T}_{new}^{\mathbb{C}}(\mathbb{1}_{forecast}) = \begin{cases} \{(t_{new_0}^{\mathbb{C}}, \boldsymbol{x}_{new_0}^{\mathbb{C}})\} & \text{if } \mathbb{1}_{forecast} = 1, \\ \{(t_{new_i}^{\mathbb{C}}, \boldsymbol{x}_{new_i}^{\mathbb{C}})\}_{i=0}^{K} & \text{if } \mathbb{1}_{forecast} = 0. \end{cases} \tag{7}
$$

For each system realization, we train in a mini-batch way by making prediction on a batch of different new trajectories.

The intractable log-likelihood in Eq. (6) can be approximated via the following evidence lower bound (ELBO):

$$
\begin{aligned}
&\log p_\theta \left( \boldsymbol{X}_{new}^{\mathbb{T}} \mid \mathcal{T}^{\mathbb{C}} \cup \mathcal{T}_{new}^{\mathbb{C}}(\mathbb{1}_{forecast}), \boldsymbol{T}_{new}^{\mathbb{T}} \right) \\
&\approx \mathbb{E}_{q(\boldsymbol{u}_{sys}|\mathcal{T}^{\mathbb{C}} \cup \mathcal{T}_{new}^{\mathbb{C}}(\mathbb{1}_{forecast}) \cup \mathcal{T}_{new}^{\mathbb{T}}) q(\boldsymbol{L}_{0_{new}}^{\mathbb{T}}|(t_0^{\mathbb{C}}, \boldsymbol{x}_{0_{new}}^{\mathbb{C}}))} \log p \left( \boldsymbol{X}_{new}^{\mathbb{T}} \mid \boldsymbol{T}_{new}^{\mathbb{T}}, \boldsymbol{u}_{sys}, \boldsymbol{L}_{0_{new}}^{\mathbb{T}} \right) \\
&\quad - \text{KL} \left[ q \left( \boldsymbol{u}_{sys} \mid \mathcal{T}^{\mathbb{C}} \cup \mathcal{T}_{new}^{\mathbb{C}}(\mathbb{1}_{forecast}) \cup \mathcal{T}_{new}^{\mathbb{T}} \right) \| q \left( \boldsymbol{u}_{sys} \mid \mathcal{T}^{\mathbb{C}} \cup \mathcal{T}_{new}^{\mathbb{C}}(\mathbb{1}_{forecast}) \right) \right] \\
&\quad - \text{KL} \left[ q \left( \boldsymbol{L}_{0_{new}}^{\mathbb{T}} \mid (t_{0_{new}}^{\mathbb{C}}, \boldsymbol{x}_{0_{new}}^{\mathbb{C}}) \right) \| p(\boldsymbol{L}_{0_{new}}^{\mathbb{T}}) \right],
\end{aligned} \tag{8}
$$

where $q \left( \boldsymbol{u}_{sys} \mid \mathcal{T}^{\mathbb{C}} \cup \mathcal{T}_{new}^{\mathbb{C}}(\mathbb{1}_{forecast}) \cup \mathcal{T}_{new}^{\mathbb{T}} \right)$ and $q \left( \boldsymbol{L}_{0_{new}}^{\mathbb{T}} \mid (t_{0_{new}}^{\mathbb{C}}, \boldsymbol{x}_{0_{new}}^{\mathbb{C}}) \right)$ has been obtained through the encoder in an amortized fashion, the prior $p(\boldsymbol{L}_{0_{new}}^{\mathbb{T}})$ is isotropic Gaussian. Appendix B derives the ELBO and Algorithm. 1 provides implementation details of the model inference procedure. Compared to Eq. (5), Eq. (8) is designed to generalize across any new trajectory within the same dynamical system. This aligns with our Bayesian Optimization (BO) setting, where we further introduce the BO algorithm in the next section.

## 4 Time Delay Constraint Process Bayesian Optimization

Through maximizing the ELBO for $\theta$, we can obtain SANODEP's predictive distribution $p_\theta(\boldsymbol{X}^{\mathbb{T}}|\mathbb{C}, \boldsymbol{T}^{\mathbb{T}}, \boldsymbol{x}_0)$ for batch of time $\boldsymbol{T}^{\mathbb{T}} = \{t_1, ..., t_N\}$ at specified initial condition $\boldsymbol{x}_0$, which is sufficient for few-shot learning tasks. In the section, specifically for our practically motivated optimization problem that incorporates one additional time delay constraint, we propose an optimization framework, leveraging SANODEP for few-shot BO in ODE and benchmark in Section. 6.2.

**Minimum Observation Delay Constraint** When optimizing Eq. (2), we assume that, while one can observe state values at any chosen time $t$ on a specific initial condition $\boldsymbol{x}_0$, as illustrated in Figure 1, the next observation can only commence after a fixed known time duration $\Delta t$. In practice, these delays stem from the need to sequentially conduct separate, smaller experiments for *state value* measurements, each requiring a known period to complete before the next can begin.

With the above context, the proposed optimization framework consists of the following two steps:

### 4.1 Initial Condition Identification

The first optimization stage identifies the optimal initial conditions necessary for starting experiments. The optimality of the initial conditions is defined as achieving the maximum expected reward after the completion of observations starting at this location. Inspired by the batch strategy to achieve a similar *non-myopic* objective (González et al., 2016; Jiang et al., 2020), we propose an **adaptive batch size** based optimization strategy for the initial condition identification:

$$
\begin{aligned}
&\text{maximize}_{\boldsymbol{x}_0 \in \mathcal{X}_0, \{t_1, t_2, ..., t_N \in \tau\}, N \in \mathcal{N}_{opt}} \ \alpha \left( \boldsymbol{x}_0, t_1, ..., t_N, p(\boldsymbol{X}^{\mathbb{T}}|\mathbb{C}, \boldsymbol{T}^{\mathbb{T}}, \boldsymbol{x}_0) \right) \\
&s.t. \ \forall i \in \{1, ..., N\} : t_i - t_{i-1} \geq \Delta t
\end{aligned}, \tag{9}
$$

where $\alpha$ is the batch acquisition function to be maximized, $\mathcal{N}_{opt} = \{1, 2, \ldots, N_{\max}\}$ is the search space for the total number of observation queries, with $N_{\max} = \lfloor (t_{\max} - t_0)/\Delta t \rfloor$. $\boldsymbol{T}^{\mathbb{T}} = \{t_1, \ldots, t_N\}$ represents the set of observation times. We omit the dependence of $p_\theta(\boldsymbol{x}|\mathbb{C}, t, \boldsymbol{x}_0)$ in $\alpha(\cdot)$ for brevity thereafter.

---

[5]For notation simplicity, we always use subscript 0 to represent the initial condition of a trajectory.

## 4.2 Choose the next query time

Once the initial condition $\boldsymbol{x}_0$ has been chosen, and our last observation query made is at time $t_n$, when still have query opportunity (i.e., $t_{max} > t_n + \Delta t$), redefine the maximum remaining trajectory observation $N_{max}$ as $\lfloor \frac{t_{max} - t_n}{\Delta t} \rfloor$, the batch size search space $\mathcal{N}_{opt}$ and time search space $\tau = [t_n + \Delta t, t_{max}]$, we choose the next query time recurrently via the following optimization problem:

$$\begin{aligned} \text{maximize}_{\{t_1, t_2, \ldots, t_N \in \tau\}, N \in \mathcal{N}_{opt}} \ & \alpha\left(\boldsymbol{x}_0, t_1, \ldots, t_N\right) \\ s.t. \ \forall i \in \{1, \ldots, N\} : t_i - t_{i-1} & \geq \Delta t \end{aligned} \tag{10}$$

**Search Space Reduction** The integer variable $N$'s search space $\mathcal{N}_{opt}$ in Eq. (9),(10), though one dimensional, can be cumbersome to optimize in practice. However, for the batch expected hypervolume improvement acquisition function (`qEHVI`) (Daulton et al., 2020) that we will use as acquisition function $\alpha$, the search space can be reduced without losing optimality:

**Proposition 1.** *For acquisition functions defined as* $\alpha() = \mathbb{E}_{p(\boldsymbol{X}^{\mathbb{T}} | \mathbb{C}, \boldsymbol{T}^{\mathbb{T}}, \boldsymbol{x}_0)} \left[ \text{HVI}\left(\boldsymbol{X}^{\mathbb{T}}, \boldsymbol{T}^{\mathbb{T}}, \mathcal{F}^* | \mathbb{C}\right) \right]$ :

$$\underset{\substack{\boldsymbol{x}_0 \in \mathcal{X}_0, \\ \{t_1, t_2, \ldots, t_N\} \in \tau, \\ N \in \{1, \ldots, N_{max}\}}}{\text{maximize}} \alpha(\boldsymbol{x}_0, t_1, \ldots, t_N) = \underset{\substack{\boldsymbol{x}_0 \in \mathcal{X}_0, \\ \{t_1, t_2, \ldots, t_N\} \in \tau, \\ N \in \{\lceil N_{max}/2 \rceil, \ldots, N_{max}\}}}{\text{maximize}} \alpha(\boldsymbol{x}_0, t_1, \ldots, t_N),$$

where HVI stands for *hypervolume improvement* based on Pareto frontier $\mathcal{F}^*$, see Definition 2 of Daulton et al. (2020). Appendix C.1 provides the proof and shows that the search space reduction property holds for generic acquisition functions that are monotonic w.r.t. set inclusions. Consequently, we define $\mathcal{N}_{opt}$ as $[\lceil \frac{N_{max}}{2} \rceil, N_{max}]$ thereafter.

---

**Algorithm 2** Model Assisted Ordinary Differential Equation Optimization Framework

1: **Input:** maximum number of experiment trajectories $N_{exp}$, minimum spacing $\Delta t$, maximum observations per trajectory $N_{max}$, initial evaluated context datasets $\mathbb{C}$, acquisition function $\alpha$, evaluated trajectory number $n = 1$, initial time lower bound $t_{lb} = t_0$, time range $\tau = [t_{lb}, t_{max}]$, observer $\boldsymbol{f}_{evolve}$ for measurement.
2: **while** $n \leq N_{exp}$ **do**
3:     $\mathcal{T} = \phi$
4:     Construct the probabilistic model for $p(\boldsymbol{x} | t, \mathbb{C}, \boldsymbol{x}_0)$
5:     # Initial Condition Optimization
6:     $\boldsymbol{x}_0^*, N^* = \arg\max_{\boldsymbol{x}_0 \in \mathcal{X}, \{t_1 = t_0, t_2, \ldots, t_N \in \tau\}, N \in [\lceil \frac{N_{max}}{2} \rceil, N_{max}]} \alpha\left(\boldsymbol{x}_0, t_1, \ldots, t_N\right)$
        s. t. $\forall i \in \{1, \ldots, N\} : t_i - t_{i-1} \geq \Delta t$
    $t_{lb} = t_0 + \Delta t, \tau := [t_{lb}, t_{max}], \mathcal{T} = \mathcal{T} \cup \{t_0, \boldsymbol{x}_0^*\}, \mathbb{C} = \mathbb{C} \cup \mathcal{T}$
7:     $N_{traj} = N_{max} - 1$
8:     # Within trajectory Next Measurement Time Scheduling
9:     **while** $N_{traj} >= 1$ **do**
10:         $N^*, t_1^* = \arg\max_{\{t_1, t_2, \ldots, t_N \in \tau\}, N \in [\lceil \frac{N_{traj}}{2} \rceil, N_{traj}]} \alpha\left(t_1, \ldots, t_N\right)$
        s. t. $\forall i \in \{1, \ldots, N\} : t_i - t_{i-1} \geq \Delta t$
11:         $\boldsymbol{x}_1^* = \boldsymbol{f}_{evolve}(\boldsymbol{x}_0^*, t_1^*)$
12:         $t_{lb} = t_1^* + \Delta t, \ \tau := [t_{lb}, t_{ub}], \ \mathbb{C} = (\mathbb{C} \backslash \mathcal{T}) \cup (\mathcal{T} \cup \{t_1, \boldsymbol{x}_1^*\}), \mathcal{T} = \mathcal{T} \cup \{t_1, \boldsymbol{x}_1^*\}$
13:         $N_{traj} = \lfloor \frac{t_{max} - t_{lb}}{\Delta t} \rfloor$
14:     **end while**
15:     $n = n + 1$
16: **end while**

---

**Optimization Framework** is outlined in Algorithm. 2. We refer to the Appendix C.1 for details of how SANODEP is utilized within `qEHVI`, together with acquisition function optimizer described in Appendix C.1. Finally, we highlight that this Bayesian Optimization framework is agnostic to the model choice of $p_\theta(\boldsymbol{X}^{\mathbb{T}} | \mathbb{C}, \boldsymbol{T}^{\mathbb{T}}, \boldsymbol{x}_0)$, as we will compare in Section. 6, both NP and GP (as a non-meta-learn model) can be used.

## 5 Related Work

**Meta-learning of Dynamical Systems**: Our work is built upon a meta-learned continuous time-based model for state prediction in dynamical systems. Singh et al. (2019) extended Neural Processes (Garnelo et al., 2018b) by incorporating temporal information from the perspective of the state-space model. However, their approach is limited to discrete-time systems. Following the NODEP framework, Dang et al. (2023) extended the model for forecasting purposes in multi-modal scenarios. Recently, Jiang et al. (2023) used meta-learning to tackle high-dimensional time series forecasting problems for generic sequential latent variable models, they leverage spatial-temporal convolution to extract the $u_{sys}$, which only works on regularly spaced time steps (e.g., image frames in their use cases). However, SANODEP extracts system dynamic's feature from multiple irregularly sampled multivariate time series using the newly proposed encoder structure. Song & Jeong (2023) explored Hamiltonian representations, which are flexible enough for cross-domain generalizations. Beyond the Bayesian meta-learning paradigm, Li et al. (2023) used classical gradient-based meta-learning (Finn et al., 2017) to meta-learn dynamics with shared properties. Auzina et al. (2024) investigated the separation and modeling of both dynamic variables that influence state evolution and static variables that correspond to invariant properties, enhancing model performance. Motivated by recent advances in extending Graph Neural Networks (GNNs) to dynamical systems, several studies have explored leveraging GNN-ODEs for modeling multi-agent interactions (e.g., Luo et al. (2025); Yuan et al. (2024)). However, such advantages is not applicable to single-agent problem setting. Additionally, several works (Luo et al., 2025; Fotiadis et al., 2023) investigate robustness to distribution shifts in system-level dynamics by separating system-level and trajectory-level dynamics encoding. While this separation improves trajectory-level prediction robustness, it is less beneficial to guard against trajectory-level predictive performance if the initial condition optimization is already misleading due to distribution shift.

**Process-Constrained Bayesian Optimization**: Vellanki et al. (2017) address process-constrained batch optimization where each batch shares identical constrained variables within a subspace. Our problem similarly treats initial conditions as constrained variables in batch optimization but differs by using a joint expected utility method instead of a greedy strategy. Additionally, our approach involves on-the-fly optimization with system evolution, leading to a novel time-constrained problem addressed by an adaptive batch BO algorithm with search space reduction. Folch et al. (2022; 2024) also consider optimization under movement (initial condition) constraints.

**Few-Shot Bayesian Optimization**: A significant amount of recent work has focused on meta-learning stochastic processes, to be used as alternatives to GP for BO. Within the (Conditional) Neural Process framework ((C)NP) (Garnelo et al., 2018a;b), Transformer Neural Processes (TNP) (Nguyen & Grover, 2022) have demonstrated robust uncertainty quantification capabilities for BO. Dutordoir et al. (2023) introduced a diffusion model-based Neural Process, providing a novel framework for modeling stochastic processes which enables joint sampling and inverse design functionality. Similar to TNP, Müller et al. (2021) approached the meta-learning problem as Bayesian inference, subsequently integrating a BO framework into this model (Müller et al., 2023). We note that while a majority of the aforementioned approaches lead to promising empirical performance on BO for regular black-box functions, their backend models (if there are any) are mainly motivated by a drop-in replacement of GP or non-temporal stochastic process. A meta-learned BO method specifically for dynamical systems is, to our best of knowledge, not revealed yet.

## 6 Experiments

This section conducts experiments on meta-learning and few-shot BO for dynamical systems. All models are implemented using `Flax` (Heek et al., 2023) and are open source, available in: `https://github.com/TsingQAQ/SANODEP`. The optimization framework is based on `Trieste` (Picheny et al., 2023).

### 6.1 Modeling Comparison

We evaluate model performance comparisons on meta-learning diverse ODE system distributions with a twofold purpose. First, we investigate whether leveraging the system information can help modeling. We also investigate whether the inductive bias could bring performance benefits.

**Baseline**: To see if adding system information helps with the modeling, we compare SANODEP against NODEP under its original testing scenario (interpolation). For the second purpose, we compare SANODEP with Neural Processes (NP) (Garnelo et al., 2018b). As mentioned in Section 5, we were not able to compare with the meta ODE models of Jiang et al. (2023) as it is not been able to be applied on irregular time series. Although GP-ODE (Heinonen et al., 2018) appears to be a promising non-meta-learning baseline for modeling dynamical systems, it is not directly applicable to our meta-testing scenario. This is due to its prohibitively expensive training process and the fact that its performance on modeling multiple irregularly sampled trajectories from the same dynamical system has not been investigated in the original work. Therefore, we instead provide results using a standard Gaussian Process with independent outputs for reference, where we use Matérn 3/2 kernel as a common choice (e.g., Xie et al. (2024); Baek et al. (2024)) and trained with maximum likelihood estimation [6]. For models that do not explicitly incorporate the ODE in their structure, we use the trajectory's initial condition augmented with time as its input. We also note that due to the context set consisting of batches of multiple trajectories sampled irregularly, prevents efficient use of scaled dot product attention-based models (e.g., Nguyen & Grover (2022)) (see Appendix B.2 for details) hence we omit for comparing these none ODE based NP models.

**Training**: Following Norcliffe et al. (2021), we treat $\mathcal{F}$ as a parametric function of a specific *kinetic model* with stochasticity induced by model parameter distributions $P$ [7]. We refer to Appendix D.1 for the model formulations, its parameter distribution $P$'s form, and the training epochs. Interpolation-only experiments use SANODEP trained using Eq. (6), setting $\lambda = 0$, and NODEP trained using Eq. (5), We refer to Appendix. B.2 for the specific training parameter setting for Algorithm. 1. For the second experimental purpose, we train all models using Eq. (6) setting $\lambda = 0$ for interpolation only training and $\lambda = 0.5$ for bi-scenario training, with 5 different random seeds for model initialization.

**Evaluation**: We evaluate the model performance on target data generated from systems sampled from the same $\mathcal{F}$, which is known as *in-distribution* generalization evaluation. Excluding GP, each model was evaluated on $10^4$ random systems, each consisting of 100 trajectories to predict in a minibatch fashion. As it is computationally infeasible to evaluate GPs on the same scale, we used a random subset of the test set, 5,000 trajectories. Appendix B.2 provides full data generation settings. Following Norcliffe et al. (2021), we assess model performance in terms of Mean-Squared Error by conducting a sequential evaluation with varying numbers of context trajectories, as illustrated in Figure 3 (and Figure 8 in Appendix D.2 for Negative-log-Likelihood). We also provide the model prediction illustration in Appendix D.3 and the averaged-context-trajectory-size performance results in Tables 2 and 3 in Appendix D.2.

Table 1: **Model comparison on interpolation tasks (mean-squared-error $\times 10^{-2}$) for a range of dynamical systems.**

| Models | Lotka-Volterra ($2d$) | Brusselator ($2d$) | Selkov ($2d$) | SIR ($3d$) | Lotka-Volterra ($3d$) | SIRD ($4d$) |
|---|---|---|---|---|---|---|
| NODEP-$\lambda = 0$ | $38.9 \pm 0.52$ | $15.5 \pm 0.379$ | $7.12 \pm 0.56$ | $367.5 \pm 58.0$ | $26.8 \pm 1.53$ | $188.1 \pm 21.8$ |
| SANODEP-$\lambda = 0$ | $\mathbf{32.1 \pm 0.55}$ | $\mathbf{8.7 \pm 1.11}$ | $\mathbf{0.52 \pm 0.03}$ | $\mathbf{254.3 \pm 60.3}$ | $\mathbf{25.1 \pm 1.85}$ | $\mathbf{137.2 \pm 12.8}$ |

**Does System Awareness Improve Predictive Performance**? Table. 1 demonstrates that SANODEP is better than NODEP in all problems, indicating the benefit of capturing system information.

**Does Incorporating Temporal Information Enhance Model Performance**? Illustrated in Figure 3, comparing with NP, except for the Lotka-Voterra problem where the SANODEP trained with mixing probability $\lambda = 0.5$ shows larger predictive variance, and Lotka-Voterra ($3d$) where NP demonstrates better performance when context trajectory numbers are small, for the rest of the problems, SANODEP is either on par or noticeably better than NP irrespective with the mixing probability. Indicating that the incorporation of right inductive bias has the potential to help modeling.

**Does Augmenting Initial Conditions Improve Modeling**? We further evaluate the performance of SANODEP by comparing models with and without initial condition augmentation, as depicted in Figure 3

---

[6]We use the GP implementation within `Trieste` and additionally follow the common numerical trick implemented in `linear_operator` (used by `GPyTorch`), where the `jitter` term is progressively increased if Cholesky decomposition fails.

[7]We discuss in Appendix E on how the flexibility of $\mathcal{F}$ may change and affect the SANODEP.

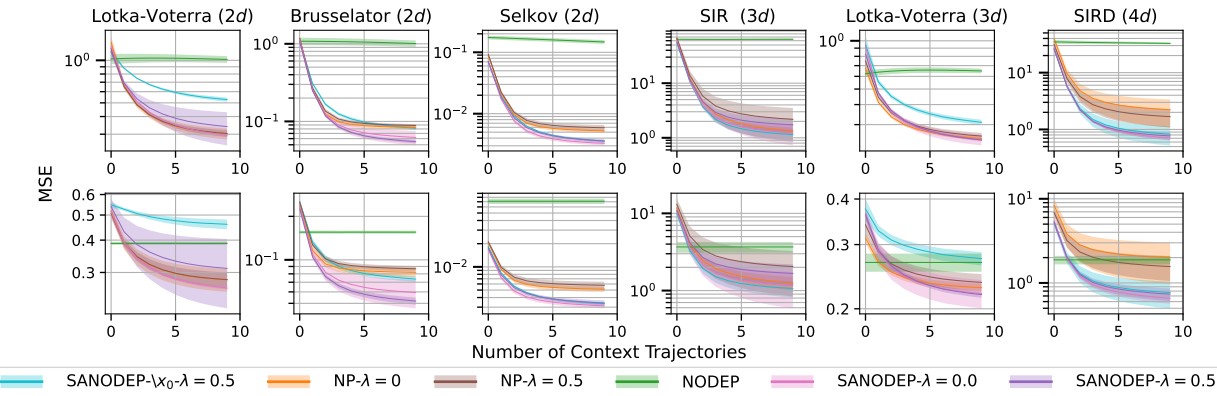

Figure 3: Model evaluation performance comparison on a different number of context trajectories. The first row corresponds to forecasting performance (prediction with only the initial condition known), and the second row represents an interpolating setting. It can be seen that SANODEP is either on-par or marginally better than NP for most problems, and the initial condition augmented encoder-based model variant provides more robust performance across problems.

(labeled as SANODEP-$\backslash x_0$ for without augmentation). The results intriguingly show that even for first order systems, models augmented with initial conditions exhibit greater robustness across various systems. This demonstrates the empirical benefits of our context aggregation approach.

**How Do Meta-Learned Models Compare with Non-Meta-Learned Models**? We provide the sequential model evaluation plot in Figure 7 including GPs in Appendix D.2. As expected, the meta-learned models show superior accuracy when the number of trajectories is small (as we have also illustrated in Figure 1), aligning with the problem setting of this study. We also remark that GP can quickly overtake when the context trajetcory number is abundant, since the Neural Process typed model is known to suffer from *under fit* when the data is abundant.

## 6.2 Few-Shot Bayesian Optimization

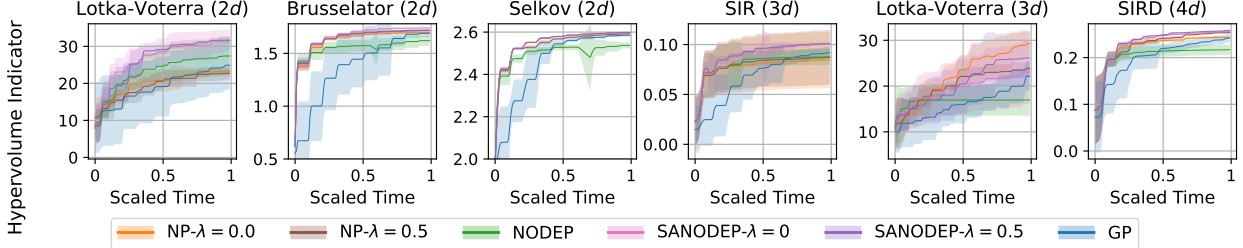

Figure 4: Comparison of few-shot BO performances in terms of mean ± standard deviation. Except for the Lotka-Voterra (3$d$) problem, SANODEP-based few-shot BO demonstrates competitive performance on all the rest of the problems.

**Baselines**: We conduct few-shot BO using the meta-learned models (SANODEP, NODEP, NP) trained in the previous section using our optimization framework (Algorithm 2). For reference, we provide GP-based BO (GP-BO) using the same framework. Since the optimization problem we consider is novel and not addressed by existing BO algorithms, we conduct our comparisons within the proposed BO framework, evaluating different models under a consistent setting.

**Optimization**: We optimize dynamical system realizations sampled from $\mathcal{F}$. The parameter settings, the objective function definitions and performance indicator calculation are provided in Table 4. For simplicity, in each optimization problem, we assume the maximum number of observations per trajectory is $\Delta t = \frac{t_{max} - t_0}{10}$.

We start with one randomly sampled trajectory with uniformly spaced observations and have an additional budget to query 10 trajectories. Each optimization is repeated 10 times with different random seeds.

**Results**: are reported in terms of the scaled experimental time versus the *hypervolume indicator*, with the reference point also provided in Table 4 (we also report the results of PI-SANODEP that will be introduced in next section.).

**Does Meta-learning Help Few-Shot Optimization in Dynamical Systems**? We highlight the merits of few-shot BO, as it consistently demonstrates significant convergence speed improvements during the early stages of BO.

**Does Dynamical System Informed Model Behaves Better than Standard Model for Optimization**? Exept for Lotka-Voterra ($3d$), SANODEP demonstrates better performance compared with Neural Processes. Again validate the potential of a deliberate consideration of temperal information.

**Does Weight Training Objective help Optimization**? For SANODEP, except in the case of the Brusselator, combining training with bi-scenario losses shows either comparable or slightly better performance than training solely in the interpolation setting. This suggests that careful design of the loss function for optimization purposes may offer modest benefits.

Finally, we provide the time profile of different models in the Appendix D.5, in general, as the involvement of simulation in inference process, SANODEP and NODEP takes more time than NP and GP when perform optimization.

## 7 Discussion

Bayesian Optimization (BO) of unknown dynamical systems is an underexplored area, with existing approaches often relying on unsuitable surrogate models. Addressing this gap, while minimising the number of costly optimization steps, we have developed a novel few-shot BO framework. Our approach extends the Neural ODE Processes (NODEP) into the System-Aware Neural ODE Processes (SANODEP), which meta-learns the system's prior information to enhance the optimization process. Through extensive benchmark experiments, we demonstrated SANODEP's potential over NODEP. Our results show that SANODEP, equipped with an optimization-driven loss function, offers competitive performance compared to other non-dynamic-aware meta-learning models, and can offer additional possibilties including parameter estimation , which can't be said with other non-dynamic-aware meta-learning models.

**Limitations**: A key limitation of SANODEP, inherited from Neural Processes, is the issue of underfitting. This necessitates a trade-off between few-shot functionality and model fitting capability. While strong prior information enhances performance in scenarios with well-defined priors, the model performance is limited by a lack of prior information about the dynamical system. Although We have discussed extensively on the variety of prior information in Appendix E, and our proposed task distribution (Appendix E.2) shows promise for cross-domain generalization, SANODEP still faces a trade-off between model fitting and prior flexibility, indicating that it cannot be used out of the box as a generic probabilistic model to optimize arbitrary dynamical system without knowing its kinetic form. Furthermore, unlike GP-BO which has established sublinear cumulative regret bounds typically due to the connection between maximum information gain and model predictive variance from the kernel (Srinivas et al., 2009), amortized latent variable models like SANODEP face several upstream open issues, including the convergence properties of amortized variational inference. Consequently, regret bounds for SANODEP based BO remain an open.

**Future investigations**: Future work will aim to develop more adaptable conditional distributions to better capture a wider range of dynamical systems. This will address the trade-off between model performance and the flexibility needed to handle diverse and complex dynamical systems without relying heavily on strong priors.

### Acknowledgments

The authors gratefully acknowledge support from BASF SE, Ludwigshafen am Rhein (BL), Engineering and Physical Sciences Research Council [grants EP/W003317/1, EP/X025292/1, EP/Y028775/1, and

EP/S023151/1] (RM, CT, JQ, BL), a BASF/RAEng Research Chair in Data-Driven Optimisation (RM), and a BASF/RAEng Senior Research Fellowship (CT).

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

# Appendix

## Table of Contents

# A  Problem Setting and Model Structures

## A.1  Problem Setting Illustrations

## A.2  Model Structure

The model structure and hyperparameters of SANODEP are summarized as follows:

- **Initial State Encoder**: $\boldsymbol{r}_{init} = \phi_{init}([t_0, \boldsymbol{x}_0]), \phi_{init} : \mathbb{R}^{d+1} \to \mathbb{R}^r$ consists of three dense layers and activation functions in between.

- **Augmented State Encoder**: $\boldsymbol{r}_i = \phi_r([t_i, \boldsymbol{x}_0, \boldsymbol{x}_i]), \phi_r := \mathbb{R}^{2d+1} \to \mathbb{R}^r$ consists of three dense layers and activation functions in between.

- **System Context Aggregation**: $\boldsymbol{r}_{sys} = \frac{1}{N} \sum_{i=1}^{N} \boldsymbol{r}_i$.

- **Context to Hidden Representation** : $\boldsymbol{h}_{sys} = \phi_{sys}(\boldsymbol{r}_{sys}), : \boldsymbol{h}_{init} = \phi_{init}(\boldsymbol{r}_{init}), \phi_{sys}(\cdot)$ and $\phi_{init}(\cdot)$ are one dense layer followed by a context encoder activation function, $\boldsymbol{h}_{sys} \in \mathbb{R}^h, \boldsymbol{h}_{init} \in \mathbb{R}^h$.

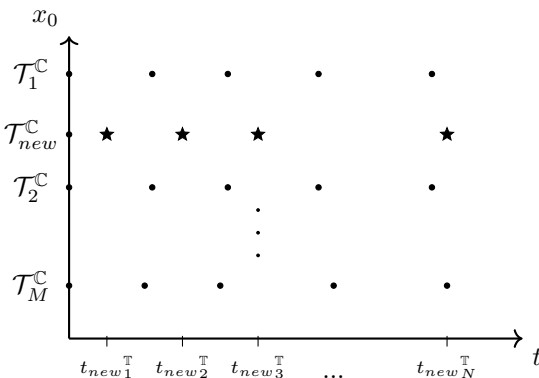

Figure 5: Illustration of the initial condition optimization problem within one state variable ($d = 1$) ODE system. The optimization involves a forecasting scenario: having observed the context set $\mathbb{C} := \left( \cup_{i=1}^{M} \mathcal{T}_i^{\mathbb{C}} \right) \cup \mathcal{T}_{new}^{\mathbb{C}}$ (including the *new* trajectory's initial condition, illustrated as •), one tries to predict the target state values $\boldsymbol{X}_{new}^{\mathbb{T}} = [\boldsymbol{x}_{new\,1}^{\mathbb{T}}, \dots, \boldsymbol{x}_{new\,N}^{\mathbb{T}}]$ at target times $\boldsymbol{T}_{new}^{\mathbb{T}} = [t_{new\,1}^{\mathbb{T}}, \dots, t_{new\,N}^{\mathbb{T}}]$ (illustrated as ★).

- **Hidden to Variational posterior of $\boldsymbol{u}_{sys}$**: $q(\boldsymbol{u}_{sys}|\cdot) = \mathcal{N}\left( \phi_{\mu_{sys}}(\boldsymbol{h}_{sys}), diag\left( \phi_{\sigma_{sys}}(\boldsymbol{h}_{sys})^2 \right) \right)$, $\phi_{\mu_{sys}} : \mathbb{R}^h \to \mathbb{R}^{d_{sys}}$ is a dense layer, $\phi_{\sigma_{sys}} := \mathbb{R}^h \to \mathbb{R}^{d_{sys}}$ is defined as $\sigma_{lb} + 0.9 * \text{softplus}(\text{Dense}(\cdot))$ , $\sigma_{lb}$ is a hyperparameter.

- **Hidden to Variational posterior of $\boldsymbol{L}_0$**: $q(\boldsymbol{L}_{0\,new}^{\mathbb{T}}|\cdot) = \mathcal{N}\left( \phi_{\mu_{init}}(\boldsymbol{h}_{init}), \phi_{\sigma_{init}}(\boldsymbol{h}_{init})^2 \right)$. $\phi_{\mu_{init}} := \mathbb{R}^h \to \mathbb{R}^l$ and $\phi_{\sigma_{init}} : \mathbb{R}^h \to \mathbb{R}^l$ has the same structure as above, but with independent weights.

- **ODE block**: $\phi_{ODE}[\boldsymbol{l}, \boldsymbol{u}_{sys}, t] \to \boldsymbol{l}'$, $\phi_{ODE} := \mathbb{R}^{d_{sys}+l+1} \to \mathbb{R}^l$, $\phi_{ODE}$ is consisted of three dense layers with nonlinear activation functions in between.

- **Decoder**: $\phi_{dec}[\boldsymbol{l}(t), \boldsymbol{u}_{sys}, t] \to \boldsymbol{x}(t)$, $\boldsymbol{l} = \mathcal{N}\left( \phi_{\mu_{dec}}[\boldsymbol{l}(t), \boldsymbol{u}_{sys}, t], \phi_{\sigma_{dec}}^2[\boldsymbol{l}(t), \boldsymbol{u}_{sys}, t] \right)$, where $\phi_{\mu_{dec}}$ and $\phi_{\sigma_{dec}}$ has the same structure as variational posterior, but with independent weights.

**Model Hyperparameters**

- Encoder $\phi_r$ output dimension $r$: 50

- Context encoder $(\phi_r, \phi_{sys}, \phi_{init})$ activation function: SiLU

- Encoder hidden dimension $h$: 50

- ODE layer $(\phi_{ODE})$ activation function: Tanh

- ODE layer hidden dimension: 50

- Decoder hidden dimension: 50

- Latent ODE state dimension $l$: 10

- Context to latent dynamics $(\phi_{\mu_{sys}}, \phi_{\sigma_{sys}})$ activation function: SiLU

- Context to latent initial $(\phi_{\mu_{init}}, \phi_{\sigma_{init}})$ condition activation function: SiLU

- Latent dynamics dimension $d_{sys}$: 45

- Variational posterior $q(\boldsymbol{L}_{0\,new}^{\mathbb{T}}|\cdot)$ variance lower bound: $\sigma_{lb} = 0.1$

- Variational posterior $q(\boldsymbol{u}_{sys}|\cdot)$ variance lower bound: $\sigma_{lb} = 0.1$

- Decoder $(\phi_{dec}, \phi_{\mu_{dec}}, \phi_{\sigma_{dec}})$ activation function: SiLU

- ODE solver: `Dopri5` with `rtol` $= 1e-5$ and `atol` $= 1e-5$

We use SiLU instead of ReLU as in the original NODEP is to enforce the differentiability, which we elaborate on in the following subsection.

### A.3 On the Differentiability of the Encoder and Decoder

Since the optimization takes the initial condition together with the time as decision variables, we need the model output to be differentiable w.r.t. these quantities. NODEP overlooked this part and utilized ReLu activation functions originally, we elaborate that the differentiability requirement practically affects our choice of activation functions.

**Time derivative** Without loss of generality, we assume we optimize a function $g$ that takes the output of the decoder $\phi_{\mu_{dec}}[\boldsymbol{l}(t), \boldsymbol{u}_{sys}, t]$ (defined in Appendix A.2) as its input [8]. The derivative of $g$ w.r.t $t$ is:

$$\frac{dg\left(\phi_{\mu_{dec}}[\boldsymbol{l}(t), \boldsymbol{u}_{sys}, t], t\right)}{dt} = \nabla_{\phi_{\mu_{dec}}} g\left(\frac{\partial \phi_{\mu_{dec}}}{\partial \boldsymbol{l}(t)} \cdot \boldsymbol{\phi}_{ODEs}[\boldsymbol{l}(t), \boldsymbol{u}_{sys}, t] + \frac{\partial \phi_{\mu_{dec}}}{\partial t}\right) + \frac{\partial g}{\partial t}, \tag{11}$$

where $\boldsymbol{\phi}_{ODEs} = \cdot$ represents the neural network structure modeling the latent ODEs. It is clear that the differentiability of model output w.r.t time $t$ needs to be enforced through $\frac{\partial \phi_{\mu_{dec}}}{\partial t}$, as long as the time $t$ is taken into account in the decoder part.

**Initial Condition Derivative** The derivative of the model output w.r.t the initial condition $\boldsymbol{x}_0$ can be represented as:

$$\begin{aligned}
&\frac{dg\left(\phi_{\mu_{dec}}[\boldsymbol{l}(t), \boldsymbol{u}_{sys}, t], t\right)}{d\boldsymbol{x}(t_0)} \\
&= \nabla_{\phi_{\mu_{dec}}} g\left(\frac{\partial \phi_{\mu_{dec}}}{\partial \boldsymbol{l}(t)} \cdot \frac{\partial \boldsymbol{l}(t)}{\partial \boldsymbol{x}(t_0)} + \frac{\partial \phi_{\mu_{dec}}}{\partial \boldsymbol{u}_{sys}} \cdot \frac{\partial \boldsymbol{u}_{sys}}{\partial \boldsymbol{x}(t_0)}\right) \\
&= \nabla_{\phi_{\mu_{dec}}} g\left[\frac{\partial \phi_{\mu_{dec}}}{\partial \boldsymbol{l}(t)} \cdot \left(\frac{\partial \boldsymbol{l}(t_0)}{\partial \boldsymbol{x}(t_0)} + \cdot\right) + \frac{\partial \phi_{\mu_{dec}}}{\partial \boldsymbol{u}_{sys}} \cdot \frac{\partial \left(\phi_{\mu_{sys}}(\boldsymbol{h}_{sys}) + diag\left(\phi_{\sigma_{sys}}(\boldsymbol{h}_{sys})\epsilon\right)\right)}{\partial \boldsymbol{x}(t_0)} \cdot\right. \\
&\left. \cdots \cdot \frac{\partial \boldsymbol{h}_{sys}}{\partial \boldsymbol{r}_{sys}} \cdot \frac{\partial \boldsymbol{r}_{sys}}{\partial \boldsymbol{x}_0}\right]
\end{aligned} \tag{12}$$

where $\boldsymbol{r}_{sys}$ and $\boldsymbol{h}_{sys}$ are defined in Appendix A.2, $\phi_{\mu_{sys}}(\boldsymbol{h}_{sys}) + diag\left(\phi_{\sigma_{sys}}(\boldsymbol{h}_{sys})\epsilon\right)$ represent the reparameterization based sampling of the latent variable $\boldsymbol{u}_{sys}$, where $\epsilon$ is a random sample from Gaussian prior. Note that the derivative of the state value w.r.t to initial condition $\frac{\partial \boldsymbol{l}(t)}{\partial \boldsymbol{x}(t_0)}$ can be obtained through the adjoint method, a derivation has been provided in Appendix C of Chen et al. (2018), without a full specification of such cumbersome terms, it is sufficient to indicate the sensibility of having differentiable activation functions in context encoding $\left(\frac{\partial \boldsymbol{r}_{sys}}{\partial \boldsymbol{x}_0}\right)$, context to latent representations $\left(\frac{\partial \boldsymbol{h}_{sys}}{\partial \boldsymbol{r}_{sys}}\right)$, as well as the decoder $\left(\frac{\partial \phi_{\mu_{dec}}}{\partial \boldsymbol{l}(t)}\right)$. For a practical calculation of the gradient of the model w.r.t decision variables in both training and optimization, we simply leverage the standard backpropagation through the ODE solver (also known as the "discretize-and-optimize" (e.g. Onken & Ruthotto (2020); Kidger (2022))) to obtain the gradient due to its speed advantage and minimum implementation effort.

In Figure 6, we also empirically show the necessity of these differentiable choices by inspecting the time and initial condition derivative with respect to $g(\cdot)$.

---

[8] we note that this is sufficient to show that the requirements of time differentiability irrespective to the consideration of $\phi_{\sigma_{dec}}[\boldsymbol{l}(t), \boldsymbol{u}_{sys}, t]$

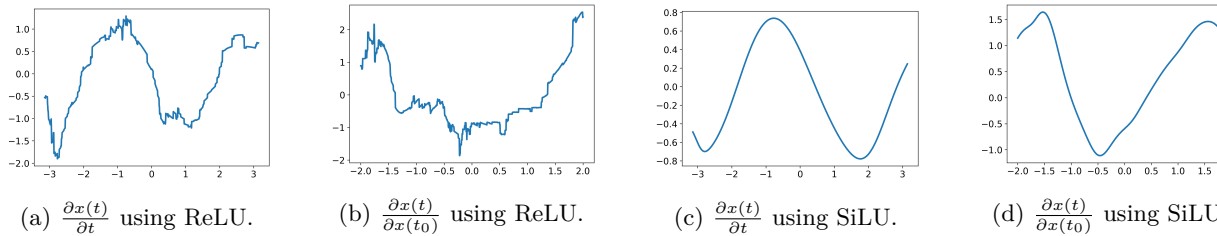

(a) $\frac{\partial x(t)}{\partial t}$ using ReLU.     (b) $\frac{\partial x(t)}{\partial x(t_0)}$ using ReLU.     (c) $\frac{\partial x(t)}{\partial t}$ using SiLU.     (d) $\frac{\partial x(t)}{\partial x(t_0)}$ using SiLU.

Figure 6: Initial condition gradient and time derivative w.r.t different activation function choices, the comparison is conducted with the same model structure but different activation functions.

## B    ELBO Derivation and Training Details

### B.1    SANODEP ELBO derivation

Note that since the forecasting scenario is a special case of interpolating scenario, below we derive through a unified representation:

$$
\begin{aligned}
&\log p\left(\boldsymbol{X}_{new}^{\mathbb{T}}|\mathcal{T}^{\mathbb{C}}\cup\mathcal{T}_{new}^{\mathbb{C}},\boldsymbol{T}_{new}^{\mathbb{T}}\right)\\
&=\log\ \int p\left(\boldsymbol{X}_{new}^{\mathbb{T}}|\boldsymbol{T}_{new}^{\mathbb{T}},\boldsymbol{u}_{sys},\boldsymbol{L}_{0_{new}}^{\mathbb{T}}\right)p\left(\boldsymbol{u}_{sys}|\mathcal{T}^{\mathbb{C}}\cup\mathcal{T}_{new}^{\mathbb{C}}\right)p\left(\boldsymbol{L}_{0_{new}}^{\mathbb{T}}\right)d\{\boldsymbol{u}_{sys},\boldsymbol{L}_{0_{new}}^{\mathbb{T}}\}\\
&=\left(\log\int p\left(\boldsymbol{u}_{sys}|\mathcal{T}^{\mathbb{C}}\cup\mathcal{T}_{new}^{\mathbb{C}}\right)p\left(\boldsymbol{L}_{0_{new}}^{\mathbb{T}}\right)\frac{q\left(\boldsymbol{u}_{sys}|\mathcal{T}^{\mathbb{C}}\cup\mathcal{T}_{new}^{\mathbb{C}}\cup\mathcal{T}_{new}^{\mathbb{T}}\right)}{q\left(\boldsymbol{u}_{sys}|\mathcal{T}^{\mathbb{C}}\cup\mathcal{T}_{new}^{\mathbb{C}}\cup\mathcal{T}_{new}^{\mathbb{T}}\right)}\right.\\
&\qquad\left.\frac{q\left(\boldsymbol{L}_{0_{new}}^{\mathbb{T}}|\left(t_0^{\mathbb{C}},\boldsymbol{x}_{\boldsymbol{0_{new}}}^{\mathbb{C}}\right)\right)}{q\left(\boldsymbol{L}_{0_{new}}^{\mathbb{T}}|\left(t_0^{\mathbb{C}},\boldsymbol{x}_{\boldsymbol{0_{new}}}^{\mathbb{C}}\right)\right)}\cdot p\left(\boldsymbol{X}_{new}^{\mathbb{T}}|\boldsymbol{T}_{new}^{\mathbb{T}},\boldsymbol{u}_{sys},\boldsymbol{L}_{0_{new}}^{\mathbb{T}}\right)d\{\boldsymbol{u}_{sys},\boldsymbol{L}_{0_{new}}^{\mathbb{T}}\}\right)\\
&\geq\mathbb{E}_{q(\boldsymbol{u}_{sys}|\mathcal{T}^{\mathbb{C}}\cup\mathcal{T}_{new}^{\mathbb{C}}\cup\mathcal{T}_{new}^{\mathbb{T}})q(\boldsymbol{L}_{0_{new}}^{\mathbb{T}}|(t_0^{\mathbb{C}},\boldsymbol{x}_{\boldsymbol{0_{new}}}^{\mathbb{C}}))}\log\left(\frac{p\left(\boldsymbol{u}_{sys}|\mathcal{T}^{\mathbb{C}}\cup\mathcal{T}_{new}^{\mathbb{C}}\right)}{q\left(\boldsymbol{u}_{sys}|\mathcal{T}^{\mathbb{C}}\cup\mathcal{T}_{new}^{\mathbb{C}}\cup\mathcal{T}_{new}^{\mathbb{T}}\right)}\right.\\
&\qquad\left.\cdot\frac{p\left(\boldsymbol{L}_{0_{new}}^{\mathbb{T}}\right)}{q\left(\boldsymbol{L}_{0_{new}}^{\mathbb{T}}|\left(t_0^{\mathbb{C}},\boldsymbol{x}_{\boldsymbol{0_{new}}}^{\mathbb{C}}\right)\right)}\cdot p\left(\boldsymbol{X}_{new}^{\mathbb{T}}|\boldsymbol{T}_{new}^{\mathbb{T}},\boldsymbol{u}_{sys},\boldsymbol{L}_{0_{new}}^{\mathbb{T}}\right)\right)\\
&\approx\mathbb{E}_{q(\boldsymbol{u}_{sys}|\mathcal{T}^{\mathbb{C}}\cup\mathcal{T}_{new}^{\mathbb{C}}\cup\mathcal{T}_{new}^{\mathbb{T}})q(\boldsymbol{L}_{0_{new}}^{\mathbb{T}}|(t_0^{\mathbb{C}},\boldsymbol{x}_{\boldsymbol{0_{new}}}^{\mathbb{C}}))}\log p\left(\boldsymbol{X}_{new}^{\mathbb{T}}|\boldsymbol{T}_{new}^{\mathbb{T}},\boldsymbol{u}_{sys},\boldsymbol{L}_{0_{new}}^{\mathbb{T}}\right)\\
&\quad-\mathrm{KL}\left[q\left(\boldsymbol{u}_{sys}|\mathcal{T}^{\mathbb{C}}\cup\mathcal{T}_{new}^{\mathbb{C}}\cup\mathcal{T}_{new}^{\mathbb{T}}\right)||q\left(\boldsymbol{u}_{sys}|\mathcal{T}^{\mathbb{C}}\cup\mathcal{T}_{new}^{\mathbb{C}}\right)\right]-\mathrm{KL}\left[q\left(\boldsymbol{L}_{0_{new}}^{\mathbb{T}}|\left(t_0^{\mathbb{C}},\boldsymbol{x}_{\boldsymbol{0_{new}}}^{\mathbb{C}}\right)\right)||p\left(\boldsymbol{L}_{0_{new}}^{\mathbb{T}}\right)\right]
\end{aligned}
\tag{13}
$$

We note the variational posterior $q(\boldsymbol{u}_{sys}|\mathcal{T}^{\mathbb{C}}\cup\mathcal{T}_{new}^{\mathbb{C}})$, $q\left(\boldsymbol{L}_{0_{new}}^{\mathbb{T}}|\left(t_0^{\mathbb{C}},\boldsymbol{x}_{0_{new}}^{\mathbb{C}}\right)\right)$ has been obtained through the encoder in an amortized approach similar in Garnelo et al. (2018b).

### B.2    Learning and Prediction Process of SANOEP

The practical learning process of SANODEP is summarized in Algorithm. 1, with parameters listed in section. F as well. In all of our subsequent experiments, we use $M_{min}=0$, $M_{max}=10$, $N_{\boldsymbol{x}_0}=100$, $N_{sys}=20$, $N_{grid}=100$, $m_{min}=1$, $m_{max}=10$, $n_{min}=0$, $n_{max}=45$. We note that all the loops have been practically implemented in batch, and all the subsampling approach has been implemented practically through `masking` in `Jax`.

**Challenges to incorporate attention** Finally, we remark that meta-learning of ODE systems is computationally challenging with models having an attention block. With lines `15-16` of Algorithm. 1, the training model takes a batch dataset of $N_{sys}\sum_{j=1}^{N_{\boldsymbol{x}0}}|\mathcal{T}j|$ points simultaneously, while the data are subsampled as context and target data separately. This is handled with a `mask`. Contemporary deep learning libraries (e.g., `Jax`) cannot leverage sparse masks to reduce attention computation complexity inherently. Hence, it is

infeasible to calculate self or cross-attention over keys with the number $\sum_{j=1}^{N_{\boldsymbol{x}_0}} |\mathcal{T}_j|$, which in our cases is tens of thousands. Generally, it is well recognized (e.g., Feng et al. (2023)) that utilizing attention blocks in neural processes with large data is computationally intensive. Additionally, lines 6 of Algorithm. 2 preferably require a dynamic update of $\mathbb{C}$ with the initial condition, which is also computationally intensive for self attention.

## C   Time Delay Constraint Optimization

### C.1   Acquisition Functions

We advocate that the framework is agnostic to the acquisition function forms as long as it can be written as expected utility form and satisfy the property mentioned in Lemma. C.1, we however mention the acquisition function form we specifically utilized for performing few-shot BO in Section 6.2 as:

$$\alpha\left(\boldsymbol{x}_0, t_1, ..., t_N, p_\theta(\boldsymbol{X}^{\mathbb{T}}|\mathbb{C}, \boldsymbol{T}^{\mathbb{T}}, \boldsymbol{x}_0)\right) \approx \frac{1}{N_{MC}} \sum_{i=1}^{N_{MC}} \text{HVI}\left(\phi_{\mu_{dec}}[\boldsymbol{l}_i(\boldsymbol{T}^{\mathbb{T}}), \boldsymbol{u}_{sys_i}, \boldsymbol{T}^{\mathbb{T}}], \boldsymbol{T}^{\mathbb{T}}, \mathcal{F}^*\right) . \tag{14}$$

In practice, we infer the current Pareto frontier $\mathcal{F}^*$ from SANODEP as well from the $M$ context trajectories $\mathcal{T}^{\mathbb{C}}$. We use $N_{MC} = 32$ as the MC sample size to approximate the expected hypervolume improvement in all our experiments.

**Acquisition Function Optimization** We note the acquisition optimizer of both initial condition (Eq. (9)), as well as optimal time scheduling (Eq. (10)) involves constraint optimization. We utilize trust region-based constraint optimization available in `Scipy`[9] to optimize acquisition function in both optimization problems starting with a uniform time scheduling (i.e., `linspace`($t_0$, $t_1$, $N$)) . Note that for the initial stage optimization, we additionally leverage a (10 instances) multi-start procedure on the initial condition decision variable only to boost the optimization performance.

### C.2   Search Space Reduction

**Proof of the search space reduction** We prove that the search space can be reduced, without eliminating the global maximum in acquisition function optimization processes (Eq. (9)).

**Lemma C.1.** *For batch acquisition function $\alpha$ defined as expected utility $\alpha = \mathbb{E}_{p(\boldsymbol{x})}(u(\boldsymbol{x}))$ where the utility function $u()$ is monotonic w.r.t set inclusion ($S \subseteq T \Rightarrow u(T) \geq u(S)$), the aquisition function is also monotonic w.r.t set inclusion: $\alpha(T) \geq \alpha(S)$.*

*Proof.* Let $T = S \cup T/S$, then $\alpha(T) - \alpha(S) = \mathbb{E}_{p(S)}\mathbb{E}_{p(T/S|S)}(u(T)) - \mathbb{E}_{p(S)}\mathbb{E}_{p(T/S|S)}(u(S)) = \mathbb{E}_{p(S)}\mathbb{E}_{p(T/S|S)}(u(T) - u(S)) \geq 0$, hence prove complete. □

**Corollary C.1.1.** *Assume that the utility function $u(\cdot)$ is monotonic with respect to the inclusion of the set in each set separately: Let $\boldsymbol{X} = \{X_1, X_2, \ldots, X_n\}$ and $\boldsymbol{Y} = \{Y_1, Y_2, \ldots, Y_n\}$ be two collections of sets, if $X_i \subseteq Y_i \ \forall i \in \{1..., n\}$ then $u(\boldsymbol{X}) \leq u(\boldsymbol{Y})$. Then, the batch acquisition function defined as $\alpha = \mathbb{E}_{p(\boldsymbol{X})}(u(\boldsymbol{X}))$ satisfies $\alpha(\boldsymbol{Y}) \geq \alpha(\boldsymbol{X})$.*

**Lemma C.2.** `qEHVI` *is a monotonic acquisition function with respect to set inclusion*

*Proof.* Using Corollary. C.1.1, Lemma C.2 and the well-acknowledged fact that the hypervolume indicator is monotonic with respect to set inclusion on each output dimensionality, the proof is straightforward. We note that the above property is asymptotically hold when `qEHVI` is monte carlo approximated. □

Finally, we start the proof of Theorem. 1:

*Proof.* when $N \in \left[1, \lceil \frac{N_{max}}{2} \rceil\right]$, $\exists i \in \left[1, \lceil \frac{N_{max}}{2} \rceil\right]$ *s.t.* $t_i - t_{i-1} \geq 2\Delta t$ meaning that one can insert a new batch point in between, forming an augmented set $T$ as input, with Lemma C.2 and Corollary. C.1.1, the proof hence complete. □

---

[9]`https://docs.scipy.org/doc/scipy/reference/optimize.minimize-trustconstr.html`

We finally remark that, with Lemma. C.1, a large series of batch improvement-based acquisition functions (e.g., parallel Expected Improvement, parallel Probability of Improvement, and parallel information-theoretic acquisition functions that can be written as expected utility form and also preserve the set inclusion monotonicity property (e.g., Qing et al. (2023))) also embrace the same search space reduction benefit.

# D Experimental Details and Additional Results

## D.1 Meta Training Data Definition

**Lotka-Voterra**

**2D Cases**

$$\frac{d\boldsymbol{x}}{dt} = \begin{bmatrix} \alpha x_1 - \beta x_1 x_2 \\ \delta x_1 x_2 - \gamma x_2 \end{bmatrix} . \tag{15}$$

The initial condition is sample from $\boldsymbol{x}(t_0) \sim U(0.1,3)^2$, the dynamics $\mathcal{F}$ are sampled from $\alpha \sim U(\frac{1}{3},1)$, $\beta \sim U(1,2)$, $\delta \sim U(0.5,1.5)$, $\gamma \sim U(0.5,1.5)$, $t_{max} = 15$. We run the meta-learn on this system distribution with 300 epochs.

**3D cases**

$$\frac{d\mathbf{x}}{dt} = \begin{bmatrix} \alpha x_1 - \beta x_1 x_2 - \epsilon x_1 x_3 \\ \delta x_2 x_1 - \gamma x_2 - \zeta x_2 x_3 \\ \eta x_3 x_2 - \theta x_3 \end{bmatrix} . \tag{16}$$

The initial condition is sampled from $\boldsymbol{x}(t_0) \sim U(0.1,3)^3$. The dynamics $\mathcal{F}$ are sampled from $\alpha \sim U\left(\frac{1}{3},1\right)$, $\beta \sim U(1,2)$, $\delta \sim U(0.5,1.5)$, $\gamma \sim U(0.5,1.5)$, $\epsilon \sim U(0.5,1.5)$, $\zeta \sim U(0.5,1.5)$, $\eta \sim U(0.5,1.5)$, and $\theta \sim U(0.5,1.5)$, with $t_{max} = 15$. We run the meta-learning algorithm on this system distribution for 300 epochs.

**Brusselator** Prigogine & Lefever (1968)

$$\frac{d\boldsymbol{x}}{dt} = \begin{bmatrix} A + x_1^2 x_2 - (B+1)x_1 \\ Bx_1 - x_1^2 x_2 \end{bmatrix} . \tag{17}$$

we leverage $A \sim U(0,1)$, $B \sim U(0.1,3)$, $\boldsymbol{x}_0 \sim U(0.1,2.0)^2$, $t_{max} = 15$. We run the meta-learning algorithm on this system distribution for 300 epochs.

**SIR model**

$$\frac{dS}{dt} = -\beta SI, \ \frac{dI}{dt} = \beta SI - \gamma I, \ \frac{dR}{dt} = \gamma I . \tag{18}$$

The initial condition is sampled from $S(t_0) \sim U(1.0,3.0)$, $I(t_0) = 0.01$, $R(t_0) = 0$, and $\beta \sim U(0.1,2)$, $\gamma \sim U(0.1,10)$. $t_{max} = 1$. We run the meta-learning algorithm on this system distribution for 300 epochs.

**SIRD model**

$$\frac{dS}{dt} = -\beta SI, \ \frac{dI}{dt} = \beta SI - \gamma I - \mu I, \ \frac{dR}{dt} = \gamma I, \ \frac{dD}{dt} = \mu I . \tag{19}$$

where $S(t_0) \sim U(10,30)$, $I(t_0) = 0.01$, $R(t_0) = D(t_0) = 0$. $t_{max} = 1$, $\beta \in [0.5,2.0]$, $\gamma \in [0.1,10]$, and $\mu \in [0.1,5.0]$. We run the meta-learn on this system distribution with 300 epochs.

**GP Vector Field**

$$\frac{d\boldsymbol{x}}{dt} = \boldsymbol{f}(\boldsymbol{x}) . \tag{20}$$

We assume a vector-valued GP as the non-parametric prior for the vector field $\boldsymbol{f} \sim \mathcal{GP}(\boldsymbol{0}, K(\boldsymbol{x}, \boldsymbol{x}'))$, with no correlations between each output as the most generic cases. To sample the vector field, we leverage the

parametric approximation of GPs through the random Fourier feature approximation of the RBF kernel Rahimi & Recht (2007) as a Bayesian linear model, which is a common approach (e.g. Qing et al. (2022)) to obtain differentiable GP samples, the vector fields in practice are generated from kernel lengthscale 0.8 and signal variance 1. The implementation also utilizes the parametric sampling approach of the `GPJax` (Pinder & Dodd, 2022) library.

## D.2 Additional Experimental Results

We report all model's performance on interpolating tasks and forecasting tasks, averaged over known trajectory ranges $M$, in Table. 2 and Table. 3.

Table 2: Model comparison (mean-squared-error $\times 10^{-2}$) for a range of dynamical systems.

| Models | Lotka-Volterra (2d) | Brusselator (2d) | Selkov (2d) | SIR (3d) | Lotka-Volterra (3d) | SIRD (4d) |
|---|---|---|---|---|---|---|
| | | | **Forecasting** | | | |
| GP | $87.2 \pm 31.9$ | $39.3 \pm 61.0$ | $31.8 \pm 6.72$ | $1769.93 \pm 980.6$ | $41.4 \pm 15.8$ | $1322.845 \pm 489.4$ |
| NODEP-$\lambda = 0$ | $102.9 \pm 6.85$ | $105.2 \pm 10.8$ | $16.1 \pm 1.14$ | $6367.784 \pm 161.8$ | $64.8 \pm 2.56$ | $3368.685 \pm 189.5$ |
| NP-$\lambda = 0.0$ | $48.6 \pm 3.1$ | $21.6 \pm 0.56$ | $1.6 \pm 0.0916$ | $952.3 \pm 73.6$ | $\mathbf{32.8 \pm 0.85}$ | $724.2 \pm 168.8$ |
| NP-$\lambda = 0.5$ | $\mathbf{47.3 \pm 1.34}$ | $22.4 \pm 1.47$ | $1.7 \pm 0.0882$ | $1057.981 \pm 207.2$ | $34.8 \pm 1.53$ | $573.0 \pm 88.3$ |
| SANODEP-$\lambda = 0$ | $48.4 \pm 2.09$ | $20.1 \pm 0.816$ | $\mathbf{1.22 \pm 0.0376}$ | $\mathbf{861.6 \pm 89.0}$ | $36.2 \pm 1.17$ | $426.4 \pm 44.0$ |
| SANODEP-$\lambda = 0.5$ | $50.6 \pm 6.4$ | $\mathbf{18.8 \pm 0.467}$ | $1.28 \pm 0.0459$ | $873.1 \pm 82.4$ | $35.5 \pm 0.824$ | $\mathbf{410.5 \pm 27.3}$ |
| | | | **Interpolating** | | | |
| GP | $52.2 \pm 20.2$ | $9.62 \pm 19.0$ | $11.7 \pm 2.76$ | $\mathbf{173.7 \pm 296.8}$ | $\mathbf{16.8 \pm 13.7}$ | $280.7 \pm 137.3$ |
| NODEP-$\lambda = 0$ | $38.9 \pm 0.517$ | $15.6 \pm 0.379$ | $7.12 \pm 0.559$ | $367.5 \pm 58.0$ | $26.8 \pm 1.53$ | $188.1 \pm 21.8$ |
| NP-$\lambda = 0.0$ | $33.2 \pm 2.59$ | $10.4 \pm 0.523$ | $0.742 \pm 0.0436$ | $286.2 \pm 29.7$ | $24.7 \pm 0.557$ | $302.0 \pm 110.9$ |
| NP-$\lambda = 0.5$ | $33.2 \pm 1.48$ | $11.0 \pm 0.477$ | $0.807 \pm 0.0638$ | $378.3 \pm 143.0$ | $25.9 \pm 1.39$ | $244.6 \pm 54.3$ |
| SANODEP-$\lambda = 0$ | $\mathbf{32.1 \pm 0.556}$ | $8.74 \pm 1.09$ | $\mathbf{0.53 \pm 0.03}$ | $257.5 \pm 60.3$ | $25.2 \pm 1.84$ | $\mathbf{139.5 \pm 13.5}$ |
| SANODEP-$\lambda = 0.5$ | $36.6 \pm 7.73$ | $\mathbf{8.06 \pm 0.265}$ | $0.573 \pm 0.0247$ | $307.8 \pm 53.5$ | $25.2 \pm 0.464$ | $143.8 \pm 14.0$ |

Table 3: Model comparison (negative log-likelihood $\times 10^2$) for a range of dynamical systems.

| Models | Lotka-Volterra (2d) | Brusselator (2d) | Selkov (2d) | SIR (3d) | Lotka-Volterra (3d) | SIRD (4d) |
|---|---|---|---|---|---|---|
| | | | **Forecasting** | | | |
| GP | $18.2 \pm 7.71$ | $30.2 \pm 25.7$ | $17.1 \pm 4.95$ | $11143.352 \pm 31169.575$ | $\mathbf{19.3 \pm 5.75}$ | $27321.033 \pm 57173.003$ |
| NODEP-$\lambda = 0$ | $119.3 \pm 21.7$ | $79.1 \pm 13.9$ | $136.4 \pm 32.1$ | $1086.616 \pm 331.3$ | $625.5 \pm 174.2$ | $447.7 \pm 86.0$ |
| NP-$\lambda = 0.0$ | $98.9 \pm 52.2$ | $\mathbf{17.7 \pm 4.41}$ | $\mathbf{4.28 \pm 1.34}$ | $\mathbf{162.6 \pm 78.6}$ | $491.2 \pm 122.1$ | $197.5 \pm 151.7$ |
| NP-$\lambda = 0.5$ | $200.9 \pm 88.0$ | $37.7 \pm 18.5$ | $5.93 \pm 0.836$ | $279.7 \pm 198.2$ | $961.1 \pm 454.1$ | $75.7 \pm 42.8$ |
| SANODEP-$\lambda = 0$ | $\mathbf{15.5 \pm 6.9}$ | $22.6 \pm 2.98$ | $7.98 \pm 1.39$ | $206.9 \pm 79.4$ | $277.5 \pm 77.9$ | $48.7 \pm 17.4$ |
| SANODEP-$\lambda = 0.5$ | $27.0 \pm 13.0$ | $23.7 \pm 4.55$ | $8.55 \pm 1.42$ | $219.4 \pm 139.7$ | $329.0 \pm 103.8$ | $\mathbf{48.3 \pm 11.0}$ |
| | | | **Interpolating** | | | |
| GP | $3.27 \pm 1.24$ | $1.88 \pm 2.0$ | $5.27 \pm 1.87$ | $718.0 \pm 2426.002$ | $\mathbf{2.69 \pm 1.61}$ | $1045.453 \pm 3397.646$ |
| NODEP-$\lambda = 0$ | $32.6 \pm 6.73$ | $12.7 \pm 2.89$ | $13.9 \pm 2.94$ | $66.4 \pm 18.2$ | $67.1 \pm 22.9$ | $22.9 \pm 8.44$ |
| NP-$\lambda = 0.0$ | $18.7 \pm 7.46$ | $\mathbf{0.254 \pm 0.61}$ | $-2.29 \pm 0.215$ | $\mathbf{22.4 \pm 10.1}$ | $65.8 \pm 11.2$ | $30.1 \pm 23.1$ |
| NP-$\lambda = 0.5$ | $34.8 \pm 14.0$ | $2.88 \pm 2.01$ | $-1.97 \pm 0.147$ | $40.1 \pm 28.4$ | $122.1 \pm 49.6$ | $12.5 \pm 6.12$ |
| SANODEP-$\lambda = 0$ | $\mathbf{2.88 \pm 1.59}$ | $1.24 \pm 0.568$ | $-1.74 \pm 0.308$ | $28.1 \pm 10.5$ | $41.3 \pm 17.7$ | $\mathbf{7.17 \pm 2.39}$ |
| SANODEP-$\lambda = 0.5$ | $6.08 \pm 3.63$ | $1.13 \pm 0.596$ | $-1.5 \pm 0.284$ | $31.3 \pm 19.3$ | $47.3 \pm 14.6$ | $7.4 \pm 1.97$ |

## D.3 Model Prediction Visual Comparison

We provide the prediction comparison of different models on a specific realization of ODE systems in Fig. 9, 12, 10, 11, 13.

## D.4 Optimization Problem Definition

The definitions of the optimization problem are provided in Table. 4.

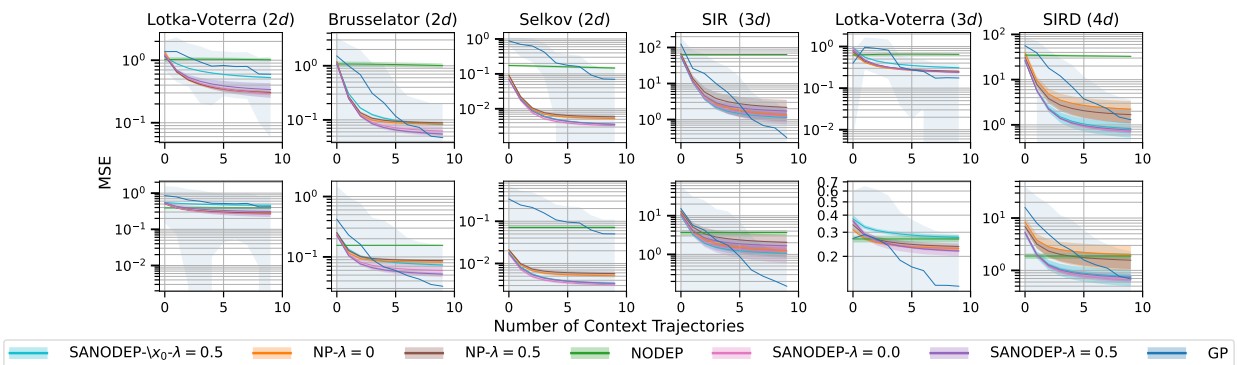

Figure 7: Sequential model evaluation performance including GPs. The first row corresponds to forecasting performance (prediction with only the initial condition known), and the second row represents an interpolating setting.

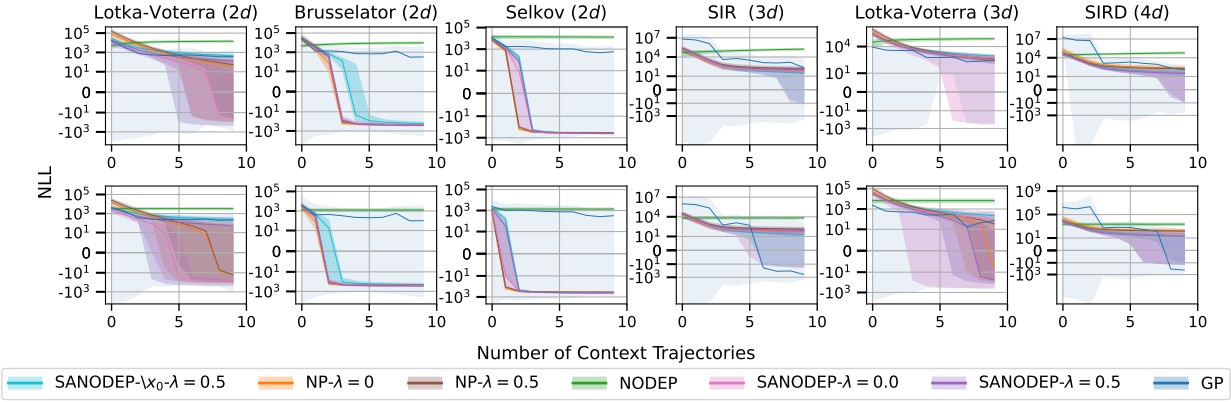

Figure 8: Sequential model evaluation performance in terms of Negative-Log-Likelihood (NLL) including GPs. The first row corresponds to forecasting performance (prediction with only the initial condition known), and the second row represents an interpolating setting.

Table 4: Optimization problem formulations

| Problem | Lotka-Volterra $(2d)$ | Brusselator $(2d)$ | Selkov $(2d)$ | SIR model $(3d)$ | Lotka-Volterra $(3d)$ | SIRD $(4d)$ |
|---|---|---|---|---|---|---|
| $g$ formulation | $x_1$ | $x_1$ | $x_2$ | $x_1/\sum_{i=1}^{3} x_i - 0.05 \sum_{i=1}^{3} x_i$ | $x_1$ | $x_1/\sum_{i=1}^{3} x_i - 0.05 \sum_{i=1}^{3} x_i$ |
| Design space | $\tau \in [0,15]$, $\boldsymbol{x}_{dec} \in [0.1, 2.0]^2$ $\boldsymbol{x}_0 = \boldsymbol{x}_{dec}$ | $\tau \in [0,15]$, $\boldsymbol{x}_{dec} \in [0.1,2]^2$ $\boldsymbol{x}_0 = \boldsymbol{x}_{dec}$ | $\tau \in [0,10]$, $\boldsymbol{x}_{dec} \in [0.1,0.5]^2$ $\boldsymbol{x}_0 = \boldsymbol{x}_{dec}$ | $\tau \in [0,1]$, $x_{dec} \in [10,30]$ $\boldsymbol{x}_0 = [x_{dec}, 0.1 \times x_{dec}, 0]$ | $\tau \in [0,15]$, $\boldsymbol{x}_{dec} \in [0.0,2]^3$ $\boldsymbol{x}_0 = \boldsymbol{x}_{dec}$ | $\tau \in [0,1]$ $x_{dec} \in [10,30]$ $\boldsymbol{x}_0 = [x_{dec}, 0.1 \times x_{dec}, 0, 0]$ |
| $\Delta t$ | 1.5 | 1.5 | 1 | 0.1 | 1.5 | 0.1 |
| Optimization ODE definition | $\alpha = 0.5$, $\beta = 1.2$, $\delta = 1.0$, $\gamma = 1.5$ | $A = 0.8$, $B = 1.5$ | $a = 0.25$, $b = 0.45$ | $\beta = 1.5$ $\gamma = 5$ | $\alpha = 0.5$ $\beta = 1.2$ $\delta = 1.0$ $\gamma = 1.5$ $\epsilon = 0.5$ $\zeta = 1.2$ $\eta = 1.0$ $\theta = 1.5$ | $\beta = 1$ $\gamma = 0.5$ $\mu = 1$ |
| Reference point | $[-1.771, 12.686]$ | $[-1.467, 3.887]$ | $[-0.474, 5.440]$ | $[0.51151, 0.79646]$ | $[-1.7557, 13.1687]$ | $[0.52198, 1.04]$ |

## D.5 Inference Time

We provide the parameter counts, training time and prediction time comparison of SANODEP vs. baselines in Table. 5, we note that all the models have the same latent dimensionality for a fair performance comparison, which is regarded as the key bottleneck of the latent variable typed meta-learning model (Kim et al., 2019).

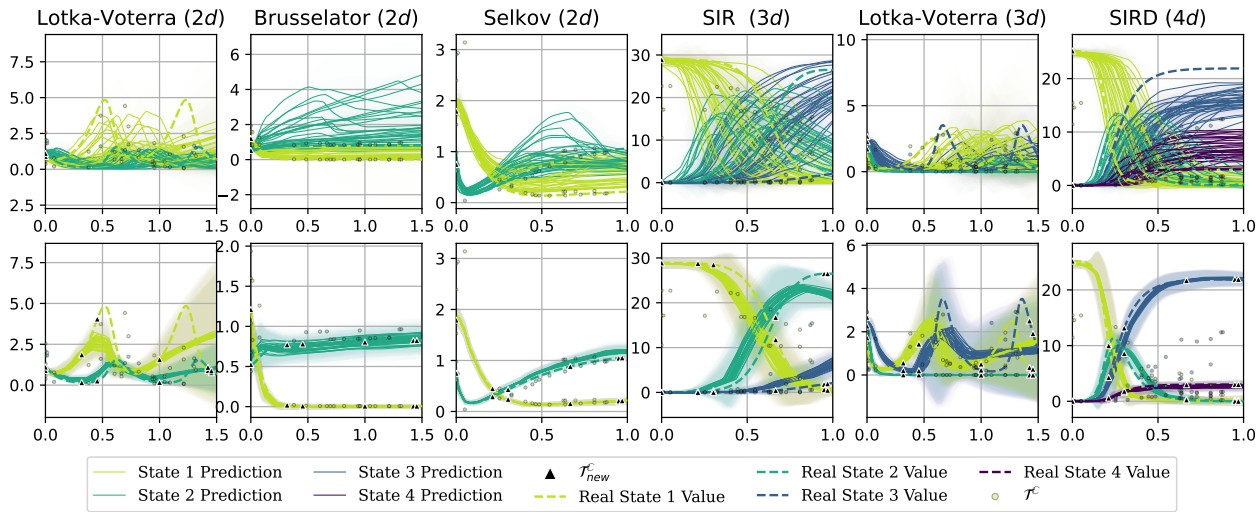

Figure 9: Neural Processes ($\lambda = 0.5$) podel merformance on test system in different meta-learning ODE problems.

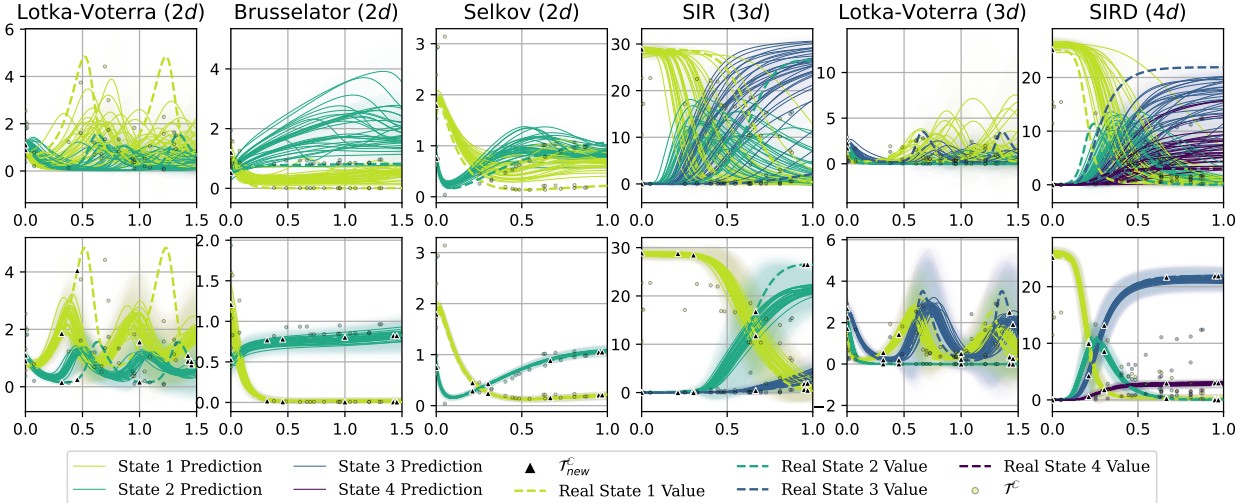

Figure 10: SANODEP-($\lambda = 0.5$) model performance on test system in different meta-learning ODE problems.

The parameter increase of SANODEP compared to NODEP is the additional encoder $\phi_r$ to handle the augmented state variable.

Table 5: Comparison of the number of parameters and prediction time of different models. We measure run time on the Lotka-Voterra ($d = 2$) problem using an NVIDIA A40 GPU, We note training time and prediction time is per mini-batch, the optimization query time is per querying one observations.

| Model | Number of Parameters | Training Time (s) | Prediction Time (s) | Optimization Query Time (s) |
|---|---|---|---|---|
| GP | NA | NA | NA | $23.81 \pm 43.94$ |
| NP | 21814 | $0.3016 \pm 0.0791$ | $0.2551 \pm 0.2337$ | $18.04 \pm 18.36$ |
| NODEP | 30286 | $0.3792 \pm 0.1892$ | $0.3421 \pm 0.2412$ | $34.71 \pm 20.62$ |
| SANODEP | 35514 | $0.3813 \pm 0.2423$ | $0.3482 \pm 0.2731$ | $39.47 \pm 37.30$ |

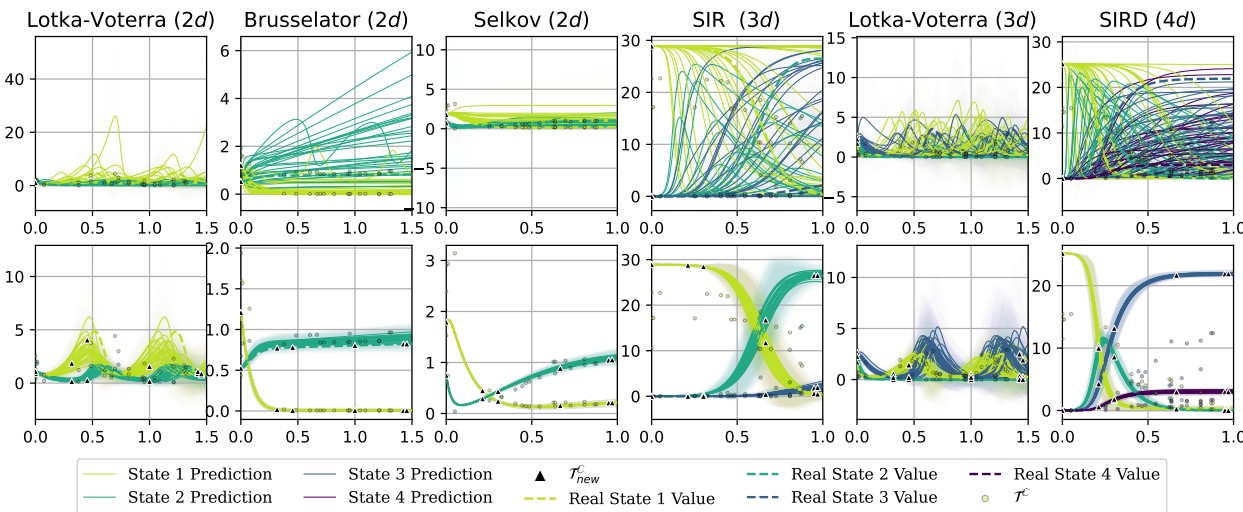

Figure 11: PI-SANODEP-($\lambda = 0.5$) model performance on test system in different meta-learning ODE problems.

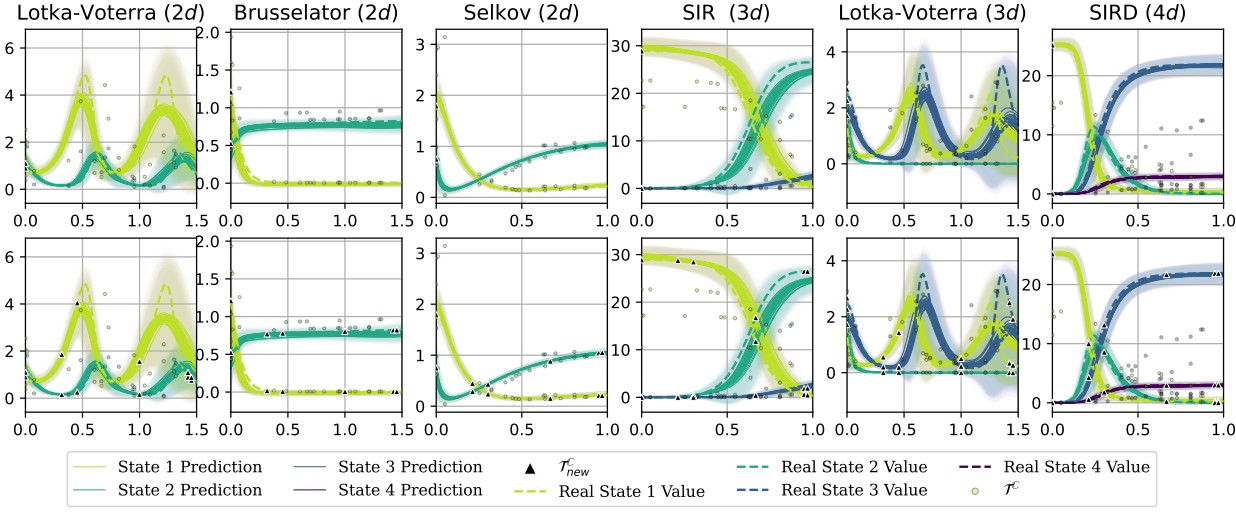

Figure 12: NODEP ($\lambda = 0$) model performance on the test system in different meta-learning ODE problems.

# E    Investigation of the prior strength, $\mathcal{F}$

## E.1    Strong Prior information: Physics-Informed SANODEP

When $\boldsymbol{f}$ is a realization of $\mathcal{F}$ and we know the parametric form of the kinetic model, we can easily integrate this kinetic form in SANODEP's model structure. We call this model variant Physical Informed (PI)-SANODEP.

As a concrete example, consider $\boldsymbol{f}$ as a realization of the Lotka-Volterra (LV) problem with unknown parameters. Under the SANODEP framework, we can replace $\boldsymbol{f}_{nn}$ in Eq. (3) with the following physics-informed form:

$$\frac{d}{dt} \begin{bmatrix} x_1 \\ x_2 \end{bmatrix} = \begin{bmatrix} \alpha_\theta x_1 - \beta_\theta x_1 x_2 \\ \delta_\theta x_1 x_2 - \gamma_\theta x_2 \end{bmatrix} \ . \tag{21}$$

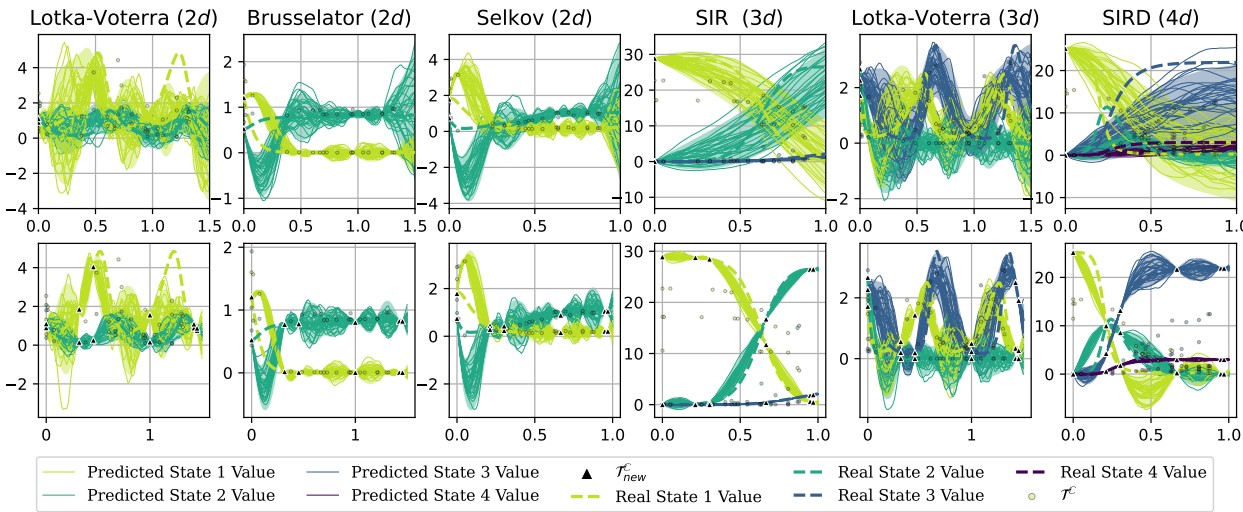

Figure 13: Performance of the GP model on the test system in different meta-learning ODE problems.

In this form, $\boldsymbol{u}_{sys} = [\alpha_\theta, \beta_\theta, \delta_\theta, \gamma_\theta]$, and $\boldsymbol{l}(t_0) = \boldsymbol{x}_0$ and $\boldsymbol{l}(t) = \boldsymbol{x}(t)$[10]. Since the parameters of the LV system are strictly positive. We use log-normal distributions for the conditional prior and variational posterior for $\boldsymbol{u}_{sys}$ in the encoder. For PI-SANODEP's training loss function $\mathcal{L}'_\theta$, since these parameters are known during meta-training, we augment our loss (Eq. (8)) by incorporating a likelihood term for these parameters, leading to the following expression:

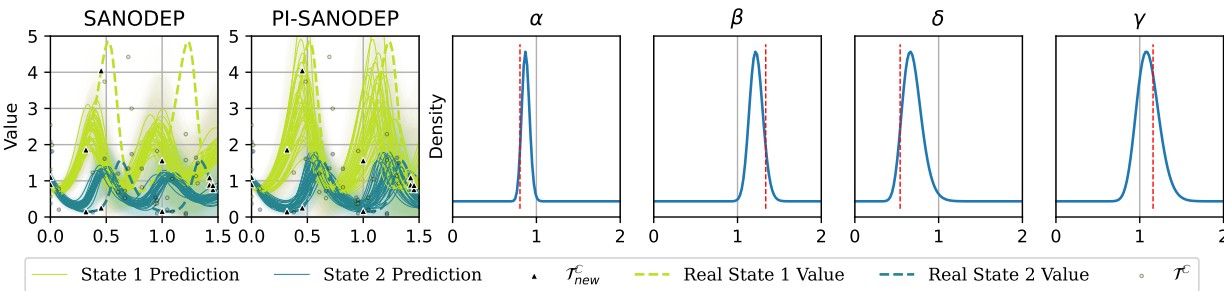

Figure 14: Model Comparison of SANODEP vs Physics Informed (PI) SANODEP, PI-SANODEP provides remarkable better prediction accuracy. In addition, PI-SANODEP provides a reasonable estimation (curve) on system parameters (dashed line).

$$\log p\left(\boldsymbol{X}^{\mathbb{T}}_{new}, \boldsymbol{u}_{sys} | \mathcal{T}^{\mathbb{C}} \cup \mathcal{T}^{\mathbb{C}}_{new}, \boldsymbol{T}^{\mathbb{T}}_{new}\right) = \log p\left(\boldsymbol{X}^{\mathbb{T}}_{new} | \boldsymbol{u}_{sys}, \mathcal{T}^{\mathbb{C}} \cup \mathcal{T}^{\mathbb{C}}_{new}, \boldsymbol{T}^{\mathbb{T}}_{new}\right) + \log p\left(\boldsymbol{u}_{sys} | \mathcal{T}^{\mathbb{C}} \cup \mathcal{T}^{\mathbb{C}}_{new}, \boldsymbol{T}^{\mathbb{T}}_{new}\right). \tag{22}$$

Since $\boldsymbol{u}_{sys}$ is known during meta-training, we use the encoder (described in Appendix A.2) as an inference network to calculate its likelihood $p\left(\boldsymbol{u}_{sys} | \mathcal{T}^{\mathbb{C}} \cup \mathcal{T}^{\mathbb{C}}_{new}, \boldsymbol{T}^{\mathbb{T}}_{new}\right) = \mathcal{N}\left(\boldsymbol{u}_{sys}; \phi_{\mu_{sys}}(\boldsymbol{h}_{sys}), diag\left(\phi_{\sigma_{sys}}(\boldsymbol{h}_{sys})^2\right)\right)$. Regarding $p\left(\boldsymbol{X}^{\mathbb{T}}_{new} | \boldsymbol{u}_{sys}, \mathcal{T}^{\mathbb{C}} \cup \mathcal{T}^{\mathbb{C}}_{new}, \boldsymbol{T}^{\mathbb{T}}_{new}\right)$, we however omit the dependence of $\boldsymbol{u}_{sys}$ and still make use of Eq. (8) to calculate. Thus, the final loss function for PI-SANODEP extends the original SANODEP loss with an added likelihood term $\log p\left(\boldsymbol{u}_{sys} | \mathcal{T}^{\mathbb{C}} \cup \mathcal{T}^{\mathbb{C}}_{new}, \boldsymbol{T}^{\mathbb{T}}_{new}\right)$, capturing both the dynamics and parameter estimation.

---

[10]In PI-SANODEP, since the ODE is explicitly defined as the kinetic form (e.g., Eq. 21) the decoder is only used to predict variance.

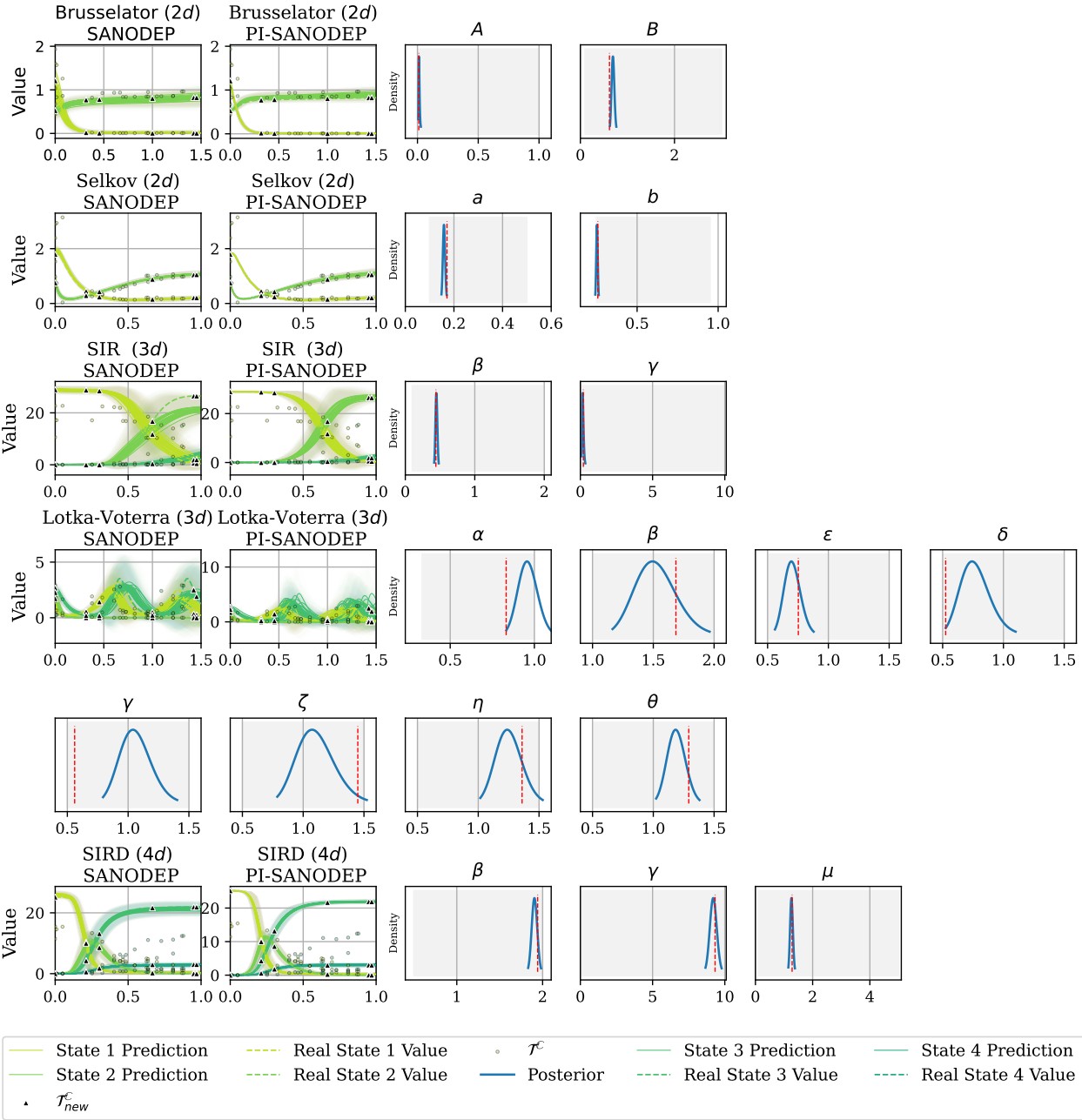

Figure 15: Additional experiments for meta-parameter estimation based on PI-SANODEP, the shaded area represents the meta-training parameter support $\mathcal{S} = \text{supp}(P)$. It can be seen that PI-SANODEP not only improves upon SANODEP over prediction (e.g., in the SIR example) but also provides a reasonable parameter estimation of the test system. Parameter estimation tends to become difficult when the number of parameters is high as can been seen from the 3-dimensional Lotka-Voterra system.

We demonstrate PI-SANODEP's performance with SANODEP on Lotka-Voterra problem in Figure 14, and on the rest of the problem in Figure 15. we note PI-SANODEP also demonstrates the potential of concurrent few-shot prediction together with parameter estimation of dynamical systems. Compared with Bayesian inference-based parameter estimation strategies (e.g. Wu & Lysy (2024)), which may take minutes to perform

estimation, PI-SANODEP shares the same advantage as *simulation-based inference* (Cranmer et al., 2020) and can estimate almost instantly since it only requires a forward pass of the encoder.

### E.2 Weaker Prior of $\mathcal{F}$: Cross-domain generalization possibility

What if we know nothing about $\mathcal{F}$? From meta-learning perspective, to still frame the problem as in-distribution evaluation, this demands an extremely flexible prior for $\mathcal{F}$ to be utilizable in the target dynamical system. To our best of knowledge, meta-learning dynamical systems to be leveraged in cross-domain scenarios is still an open direction. Certain approaches have resorted to incorporating additional dynamical system properties (e.g., energy conserving system Song & Jeong (2023)) which limit the scope of system types. We propose and leverage a vector valued GP prior for $\mathcal{F}$ as a flexible task distribution.

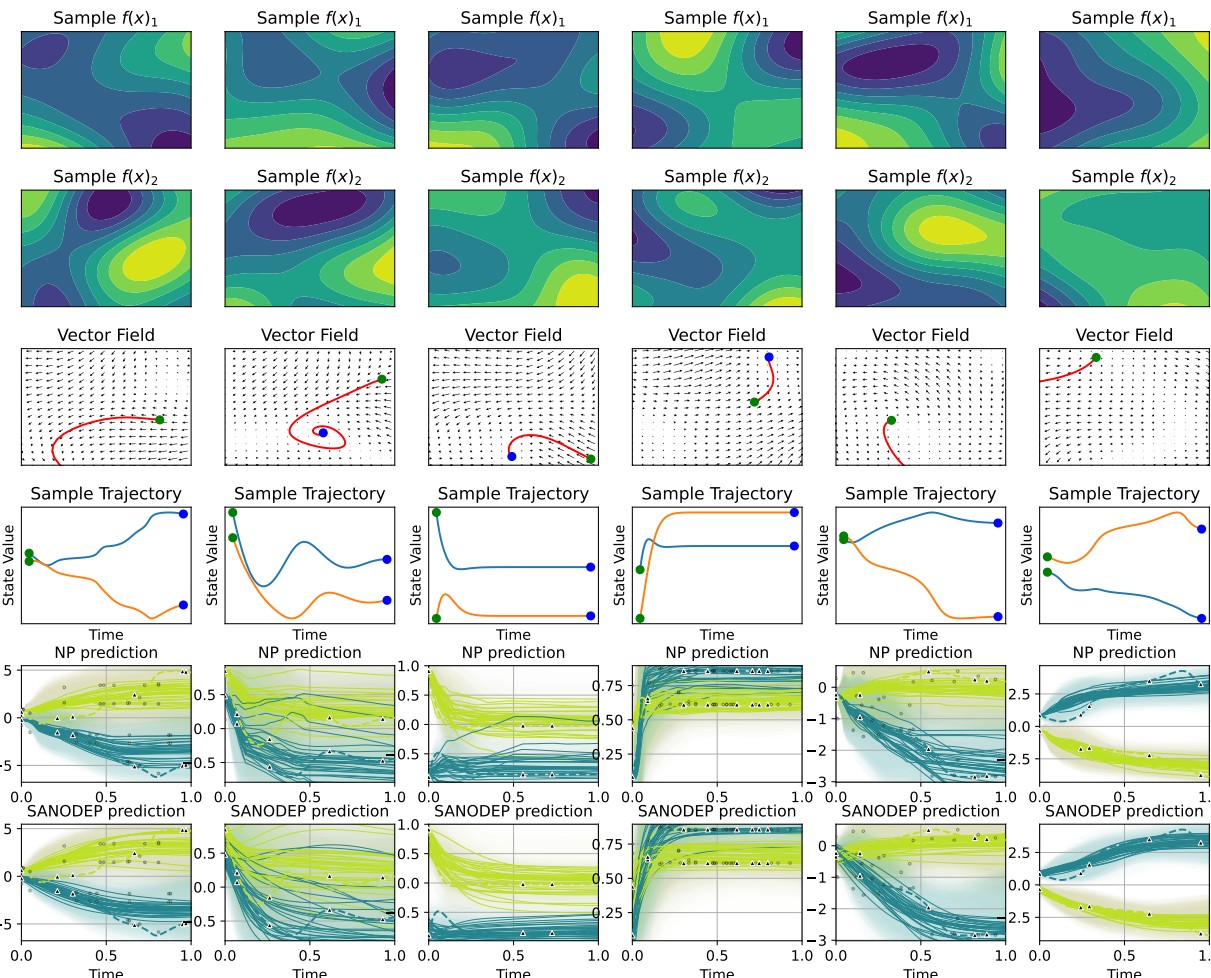

Figure 16: Task distribution generated by vector-valued Gaussian Process (GP) priors. The first two rows display samples from parametric GP priors. The third and fourth rows show the sampled vector fields and the resulting trajectories starting from sampled initial conditions. The last two rows depict the meta-model predictions on these trajectories. It can be seen that while the GP-based vector field has introduced a very flexible task distribution, both meta-learn models tend to underfit on these trajectories.

Inspired by the recent approach of using Gaussian Processes (GPs) to model vector fields (e.g., Heinonen et al. (2018)), we propose to use vector-valued GP as the non-parametric prior for the vector field, denoted by $\boldsymbol{f} \sim \mathcal{GP}(\mathbf{0}, K(\boldsymbol{x}, \boldsymbol{x}'))$, as task distributions for meta-learning for dynamical systems. Further details on the data used for meta-training are provided in Appendix D.1. The dynamical systems sampled from this

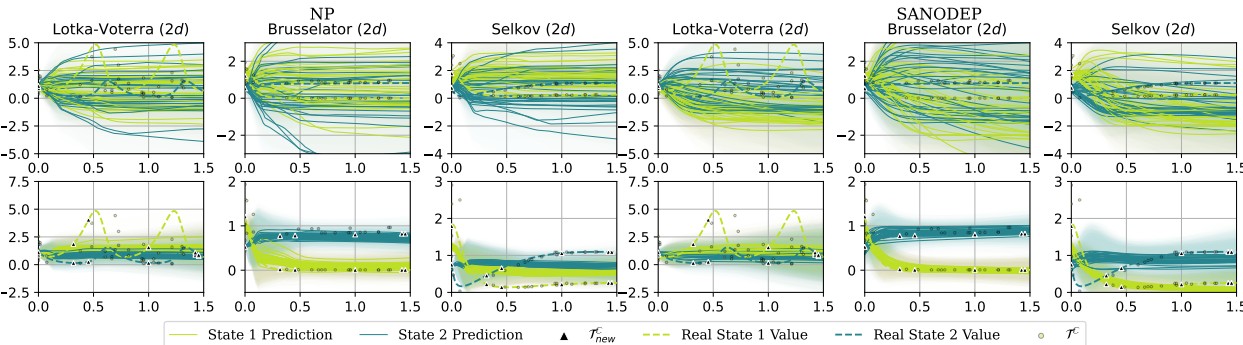

Figure 17: Cross domain generalization exploration: SANODEP and NP meta trained on GP-ODE seems to be able to capture the trajectories from Bruseelator system.

vector-valued GP prior, as illustrated in Figure 16, demonstrate its ability to represent a broad distribution of dynamical systems. Additionally, we offer a visual comparison of SANODEP and NP on the sampled trajectories. This comparison reveals a trade-off between fitting accuracy and trajectory flexibility for both models. Specifically, both models tend to significantly underfit trajectories that involve oscillations, indicating the noticable complicity of this meta-learning problem.

We also explore whether this flexible task distribution facilitates cross-domain generalization. To this end, we employ the trained model on different dynamical systems that share the same state variable dimensionality, as shown in Figure 17. Both meta-learned models demonstrate some potential for cross-domain generalization particularly from the Brusselator system. However, generalization to systems involving oscillatory behaviors proves to be especially challenging.

# F  Notations

Table 6: Nomenclature Table

| Notation | Meaning |
|---|---|
| $\boldsymbol{x}$ | State variable vector |
| $d_{\boldsymbol{x}}$ | Dimension of the state space |
| $t_0$ | Initial time for the dynamical system |
| $t_i$ | Trajectory time samples, sampled irregularly |
| $t$ | Termination time for a trajectory |
| $\mathbb{C}$ | Context Sets |
| $\boldsymbol{f}$ | Vector field governing the dynamics |
| $g(\cdot)$ | Aggregation function used in optimization |
| $L_0$ | Latent initial condition |
| $L_D$ | Dynamics of the system |
| $\phi_r$ | Context encoding |
| $\boldsymbol{r}$ | Context representation vector |
| $\boldsymbol{l}_0$ | Realizations of latent initial conditions |
| $\boldsymbol{l}(t)$ | Latent states at time $t$ |
| $\boldsymbol{u}(\boldsymbol{u}_{sys})$ | Realizations of the control term representing latent dynamics |
| $\boldsymbol{f}_{nn}$ | Neural network parameterized vector field |
| $\theta_{ode}$ | Parameterisation of the ODE system |
| $\boldsymbol{f}_{evolve}$ | Numerical solver implementation of the vector field |
| $h$ | Instantenous time variable in continuous dynamics |
| $d_{\boldsymbol{l}}$ | Dimension of the latent state space |
| $\mathbb{T}$ | Target Sets |
| $\boldsymbol{T}^{\mathbb{T}}$ | Target times for predictions |
| $\boldsymbol{X}^{\mathbb{T}}$ | Target state values |
| $\mathcal{F}$ | Distribution of dynamical systems |
| $P$ | Parametric Task distribution, often stochastic |
| $\mathcal{X}_{\boldsymbol{x}}$ | State variable space |
| $\mathcal{X}_0$ | Initial condition space |
| $\tau$ | Time domain for the system evolution |
| $M$ | Number of context trajectories |
| $N_l$ | Number of context elements in trajectory $l$ |
| $\mathcal{T}^{\mathbb{C}}$ | Context observations from $M$ trajectories |
| $\mathcal{T}_l^{\mathbb{C}}$ | Context observations from the $l$th trajectory |
| $\mathcal{T}_{new}$ | A new trajectory which augments the context set $\mathbb{C}$ |
| $\mathcal{T}_{new}^{\mathbb{C}}$ | Context observations from the new trajectory |
| $\mathcal{T}_{new}^{\mathbb{T}}$ | Target observations from the new trajectory |
| $D_{sys}$ | Latent variable conditioned on $M+1$ trajectories |
| $\boldsymbol{u}_{sys}$ | Realization of system-aware latent dynamics |
| $\boldsymbol{r}_{sys}$ | System context representation vector |
| $\phi_{r_{sys}}$ | Context encoding augmented with the initial state |
| $\boldsymbol{X}_{new}^{\mathbb{T}}$ | Target state values for the new trajectory |
| $\boldsymbol{T}_{new}^{\mathbb{T}}$ | Target times for the new trajectory |
| $\boldsymbol{x}_{new_i}^{\mathbb{C}}$ | State vector samples from the new trajectory |
| $t_{new_i}^{\mathbb{C}}$ | Time samples from the new trajectory |
| $K$ | Number of observations in the new trajectory |
| $\mathcal{L}_\theta$ | SANODEP's multi scenario loss function parametrised by $\theta$ |
| $p_\theta$ | SANODEP prediction parametrised by $\theta$ |
| $\mathbb{1}_{forecast}$ | Bernoulli Indicator |
| $\lambda$ | Parameter for Bernoulli distribution in decision-making process |

Table 6: Nomenclature Table (Continued)

| Notation | Meaning |
|---|---|
| $\mathcal{N}_{opt}$ | Search space of the number of observations in the trajectory |
| $\Delta t$ | Minimum delay between observations |
| $t_{max}$ | Maximum time observation |
| $N_{max}$ | Maximum number of observations per trajectory |
| $\alpha(\cdot)$ | Batch acquisition function |
| HVI | Hypervolume improvement based on the Pareto frontier |
| $\mathcal{F}*$ | Pareto frontier |
| $M_{min}$ | Minimum number of context trajectories observed in a system |
| $M_{max}$ | Maximum number of context trajectories observed in a system |
| $N_{\boldsymbol{x}_0}$ | Sampled number of initial conditions within a system |
| $N_{sys}$ | Number of dynamical system samples per training iteration |
| $N_{grid}$ | Number of points in the time grid for trajectory evaluation |
| $m_{min}, m_{max}$ | Min and max context points within a trajectory |
| $n_{min}, n_{max}$ | Min and max target points beyond context in a trajectory |
| $\mathcal{S}$ | The support set of the stochastic parameters for meta-task distribution |
| $\boldsymbol{T}_{grid}$ | Time grid used for trajectory evaluations |

