# OpenReview forum: "System-Aware Neural ODE Processes for Few-Shot Bayesian Optimization"
_TMLR — Accepted by TMLR_

### Review · Reviewer_jc5T · 2024-12-04

**Summary Of Contributions:**

The paper proposes an extension of the NODEP framework towards dealing with multiple trajectories. The multiple trajectories are constructed by realization of different initial conditions.
The novel system is then used as proxy model in a Bayesian Optimization problem.  The authors frame the optimization problem as a two step problem, where first the initial condition is optimized and second the termination time is optimized.
Similar optimization problems may be found in real world production systems.

**Audience:**

Yes

**Broader Impact Concerns:**

Not applicable.

**Claims And Evidence:**

Yes

**Requested Changes:**

Clarifying difference to NODEP:
-  Multiple Trajectories:  Is it possible deal with different initial conditions by increasing the feature dimension of $x$, i.e, replacing $\phi=(t_i, x_i)$ with $\phi=(t_i, z_i)$ with $z_i = [x_0, x_i]$ and treating it as one dataset? Similar as augmented neural odes are constructed.
-  Loss: As the loss function is different in comparison to the original NODEP paper: Can you discuss the influence and emperically demonstrate the influence of each term in Eq. 6?

GP Baseline:
- Fig 4. suggests that the proposed framework is able to outperform the GP baseline. However, the authors provide little information about how the GP was constructed. Beyond that, the authors omit comparing against GP-ODEs, which was though cited.
See for example Heinonen 2018 and subsequent work for this. I think a comparison against this model class is crucial.

Minors (Notation):
- Section 2.2 --> notation switching $d$ vs $u$.
- what is the benefit of having $f_{evolve}$ if you can just write $l(t)$?

**Strengths And Weaknesses:**

Strengths:
- The problem of finding the optimal initial conditions of unkown dynamical system is relevant.
- The paper has a detailed related work section.

Weakness:
- The contribution is not clearly defined, as the change to NODEP framework seems minor. See Requested Changes in order to make the contribution clearer.
- Fig 1. is hard to grasp and does not show the benefit of the model: difference filled vs blank red dots? GP seems to do as good?
- The notation is partially confusing (See "Minor" und requested changes)
- While the paper is overall well written, Sec. 4 is hard to follow.
- (PI)-SANODEP seems orthogonal to the rest of the papers story.

---

> ### Author Response · Authors · 2024-12-19
>
> We thank the reviewer for the detailed and constructive feedback and have addressed the typo suggestions in the updated paper.
>
> > Fig. 1 is hard to grasp and does not show the benefit of the model: the difference filled vs blank red dots? GP seems to do as good?
>
> Fig. 1 illustrates the performance of few-shot Bayesian Optimization (BO) across different models. Specifically:
> - The goal is bi-objective optimization: maximize the objective function (contour levels) while minimizing integration time.
> - Red dots represent sampled points: blank dots are initial randomly sampled trajectories, and filled dots indicate BO-selected samples.
> - The main cost in optimization is the switching of different initial conditions. The key benefit is that SANODEP achieves the optimal initial condition (at $\boldsymbol{x}_0=0$) with one additional trajectory, highlighting superior optimization efficiency compared to other models, including GP.
> To improve clarity, we have revised the caption for Fig. 1 in the updated paper.
>
> > Multiple Trajectories: Is it possible to deal with different initial conditions by increasing the feature dimension of $\mathbf{x}$, i.e., replacing $\phi = (t_i, \mathbf{x}_i)$ with $\phi = (t_i, \mathbf{z}_i)$ where $\mathbf{z}_i = [\mathbf{x}_0, \mathbf{x}_i]$ and treating it as one dataset? Similar to how augmented neural ODEs are constructed.
>
> If we understand correctly, the reviewer refers to the approach in [1], which augments the state $z(t):=x(t)$ with additional state variables $h(t)$ to address topology-induced expressivity issues. However, we believe this method is orthogonal to the feature extraction approach discussed in our work. Our method achieves permutation-invariant representation by formulating a single dataset where each data point is augmented with its corresponding initial condition.
>
> If a different reference was intended, we are happy to provide further clarification.
>
> > The contribution is not clearly defined
>
> The main contribution of our work is the extension of NODEP for few-shot BO, motivated by the optimization problem. This involves:
> a novel prediction setting across multiple trajectories, which necessitates the use of set-based representations for multiple context trajectories in NODEP. Additionally, we developed a BO framework tailored to this problem and conducted benchmarking within this new problem setting. We also believe that BO for dynamical systems provides a novel problem setting.
>
> > (PI)-SANODEP seems orthogonal to the rest of the paper's story.
>
> Our primary focus is on demonstrating the utility of handling varying prior strengths, which we believe is a relevant topic in meta-learning-based modeling and task-specific model design hence provided as a discussion.
>
> > discuss the influence and empirically demonstrate the influence of each term in Eq.~6?
>
> We note starting from Eq.~6, one can recover the loss of the original NODEP as:
> $\mathcal L_{\theta}  =  \mathbb E_{\boldsymbol f \sim \mathcal F, \mathcal T_{new}^{\mathbb T}, \mathcal T_{new}^{\mathbb C}} \text{log}\ p_{\theta} \left({{\boldsymbol X_{new}^{\mathbb T}}} | \mathcal T_{new}^{\mathbb C}, {\boldsymbol T_{{new}}^{\mathbb T }}\right)$
>
> with $\mathbb 1_{forecast}=0$ and diminishing $M$, $\mathcal{T}^\mathbb{C}$, as these represents context trajectories from different initial conditions.
>
> The additional terms on top of this loss form to reach our loss are for two purposes:
> - Augmented Context Data: SANODEP leverages trajectories from multiple initial conditions ($\mathcal{T}^{\mathbb{C}}$ from $M$ trajectories) to infer the system property. Section 6.1 empirically demonstrates its superior performance over NODEP trained on a single-trajectory loss.
> - Forecasting Indicator $\mathbb 1_{forecast}=0$: This term allows tailoring the loss for either forecasting (initial condition optimization, Eq. 9) or interpolation (trajectory observation scheduling, Eq. 10). Section 6 shows that incorporating forecasting loss marginally improves optimization performance.
> - Empirically, we present SANODEP with $\mathbb 1_{forecast}=0$ (which is similar to the original NODEP training setting), and $\mathbb 1_{forecast}=0.5$, which focuses more on pure forecasting scenario for the sake of initial condition optimization. and discussed in "Does Weight Training Objective Help Optimization" in section 6 that shifting from a pure interpolating loss (the original NODEP loss setting) marginally benefits the optimization problem.

---

> > ### Author Response · Authors · 2024-12-19
> > **Continue response**
> >
> > > GP baseline details
> >
> >  We used a standard GP with ARD-enabled Matérn-3/2 kernels, without assuming output correlation and trained via maximum a posteriori. This detail will be added to Section 6.
> >
> > > Compare with GP-ODE
> >
> > We agree that GP-ODE [2] is relevant for modeling dynamical systems though as a non-meta-learned model. Upon further evaluation of the feasibility of performing a meaningful comparison, we however identified several challenges:
> > - **Time Complexity**: Training the original GP-ODE [2] is extremely time-consuming, and benchmarking it in a meta-test setting is non-trivial. This would require evaluating model performance across many different tasks (e.g., training and testing 5000 separate GP-ODE models for varying numbers of context trajectories and system realizations, each taking half to even several hour to train) is beyond the capacity of available computational resources within an affordable time.
> >
> > - **Implementation Challenges with Multiple-irregularly sampled trajectories**: While the follow-up multiple-shooting version of GP-ODE [3] is significantly more efficient than the original, it was originally introduced and implemented [4] to handle single trajectories. Extending its implementation to handle irregularly sampled time steps across multiple trajectories introduces non-trivial complexity, particularly in parallel integration for each shooting interval and proper handling for meta-testing purposes.
> >
> > **None-meta learn model comparison is off-focus** Additionally, our primary claim is not that SANODEP is the best model for generic dynamical systems modeling and optimization, which itself is not investigated and it is not sensible to compare meta-learn with non-meta-learn models. Our main claim is that SANODEP provides a better inductive bias compared to Neural Processes (NP) and is preferable for few-shot BO in dynamical systems. While a comparison with GP-ODE could offer insights into its empirical performance on modeling multiple irregularly sampled trajectories in a non-meta-learned setting, and potentially broaden the scope of the paper to generic dynamical systems optimization, such an analysis would deviate from our primary focus on few-shot BO and require extensive adaptations that may be of independent research interest. If the reviewer believes that a specific aspect of such a comparison is still necessary under this paper's scope, we would be happy to allocate additional time to conduct an extended extensive evaluation upon further elaboration.
> >
> > [1] https://arxiv.org/abs/1904.01681
> > [2] https://proceedings.mlr.press/v80/heinonen18a.html?ref=https://githubhelp.com
> > [3] https://arxiv.org/abs/2106.10905
> > [4] https://github.com/hegdepashupati/gaussian-process-odes
> >
> > We thank the reviewer again for the valuable comments and have included the aforementioned explanations in the updated paper. We would like to clarify any aspects of unaddressed concerns upon further requests.

---

> > > ### Author Response · Authors · 2025-01-11
> > >
> > > We thank the reviewer again for the valuable comments. In response, we have revised the paper with all changes highlighted in blue text including:
> > >
> > > - improving the clarity of Figure 1's caption.
> > > - enhancing the clarity of the optimization section.
> > > - reorganizing the PI-SANODEP section into the appendix as a complementary discussion.
> > > - explaining the reasons for not comparing with GP-ODE.
> > >
> > > We are happy to address any additional concerns or provide further clarifications if there are any.

---

> > > > ### Author Response · Authors · 2025-02-05
> > > >
> > > > Dear Reviewer jc5T,
> > > >
> > > > We hope this message finds you well.
> > > >
> > > > We wanted to follow up regarding the responses we submitted to address your comments. As some time has passed, we kindly ask if you’ve had the opportunity to review them and could confirm whether our responses have adequately addressed your concerns. If any further clarification or additional information is needed, we would be happy to provide it.
> > > >
> > > > Thank you very much for your time and consideration.
> > > >
> > > > Best regards,
> > > > The Authors

---

> > > > > ### Comment · Reviewer_jc5T · 2025-02-07
> > > > > **Response**
> > > > >
> > > > > I am overall fine with the clarifications. However, the focus of the paper is rather narrow as "None-meta learn model comparison is off-focus" and consequently comparisons to well established baselines are missing such as  [1] or extensions of the GP setup.

---

> > > > > > ### Author Response · Authors · 2025-02-07
> > > > > >
> > > > > > We sincerely appreciate the reviewer’s prompt response and insightful comments. As highlighted in contribution point 3, our primary focus is on comparing meta-learned models. Since our approach does not introduce a standalone mechanism to enhance model flexibility beyond the meta-learning framework—unlike [1], which we leverage within our augmented latent ODE—we believe that a direct comparison with [1] may not be necessary within the current scope of our work.
> > > > > >
> > > > > > Regarding the broader impact on dynamical system modeling, our contribution lies in developing a meta-learned model specifically designed for optimising initial conditions and termination times through a meta-learned few-shot Bayesian Optimisation framework, rather than proposing a general-purpose model for dynamical system modeling. If the distinction between these objectives requires further clarification, we would be happy to incorporate additional discussion in the introduction or related work section.
> > > > > >
> > > > > > [1] https://arxiv.org/abs/1904.01681

---

### Review · Reviewer_XihC · 2024-12-08

**Summary Of Contributions:**

The paper presents a meta-learning approach for few-shot bayesian optimization in dynamical systems, which addresses the challenges of optimizing initial conditions and termination times by leveraging prior system information.

**Audience:**

Yes

**Claims And Evidence:**

Yes

**Requested Changes:**

See Weaknesses.

**Strengths And Weaknesses:**

Strengths
- The paper effectively addresses the practical challenge of optimizing dynamical systems with limited evaluative resources.
- The proposed SANODEP method is innovative in its integration of system-awareness into the Neural ODE Processes.
- The experimental section is thorough, with a variety of dynamical systems tested, demonstrating the robustness of SANODEP across different scenarios.

Weaknesses
- How does the SANODEP framework handle dynamical systems with highly nonlinear or complex interactions that may not be well-captured by the assumed ODE forms?
- How sensitive is SANODEP to the choice of hyperparameters, particularly those related to the context embedding block and the bi-scenario loss function?
- What are the theoretical guarantees for the convergence of the proposed two-stage Bayesian Optimization framework, especially when dealing with non-convex objective functions common in dynamical systems?
- What is the robustness of SANODEP against adversarial attacks or perturbations in the input data？
- There are several relevant methods missing in the manuscript that need to be discussed and analyzed: 'PGODE: Towards High-quality System Dynamics Modeling' and 'EGODE: An Event-attended Graph ODE Framework for Modeling Rigid Dynamics'.
- Can the authors discuss the theoretical implications of the proposed method on the exploration-exploitation trade-off in the context of Bayesian Optimization, and how this might influence the optimization of initial conditions and observation timings?

---

> ### Author Response · Authors · 2024-12-19
>
> We thank the reviewer for their thoughtful feedback and insightful questions.
>
> >Exploration-Exploitation Trade-off and Convergence Guarantees
>
> The initial condition optimization involves a trade-off between selecting an initial condition that is highly likely to provide good overall rewards after sampling and one that is more uncertain but has the potential to achieve better rewards.
>
> Establishing a regret proof for this optimization problem presents the following challenges:
> - Most contemporary regret proofs in BO assume objective functions drawn from a GP or with a low RKHS norm [1]. However, the property of objective function distribution in dynamical systems remains an open research question.
> - Meta-BO regret proofs (e.g., [2]) rely on GP models and the diminishing epistemic uncertainty assumption, which does not consistently hold for Neural Process-based meta-learn models.
> - The backbone acquisition function (EHVI) lacks regret proof itself even on GPs.
>
> Hence, a convergence guarantee for the two-stage BO framework in a dynamical system-based objective function has not yet been established. We have included an explanation in the updated paper as well.
>
> > Missing refs of PGODE and EGODE and dealing with out-of-distribution setting?
>
> We thank the reviewer for mentioning these relevant references. We have included them with a discussion in the updated paper and elaborate here.
>
> Regarding model structure, Graph Neural Networks (GNNs) have demonstrated strong performance in modeling interacting systems, particularly in multi-agent scenarios [3]. Both PGODE and EGODE leverage GNNs to represent dynamical systems effectively. However, the advantages of utilizing GNNs for single-agent dynamical systems remain relatively underexplored and may not directly align with our optimization-focused objectives.
>
> As for the robustness in out-of-distribution (OOD) settings, PGODE encodes system-level and object-level (or agent-level) dynamics separately, using an adversarial training framework to enhance robustness against OOD shifts in system-level dynamics. Similarly, disentanglement methods, as explored in [4], aim to separate trajectory-level information from system-level dynamics to produce robust predictive distributions. These approaches are well-suited for tasks that involve detailed trajectory/objective level predictions that require robust handling of OOD conditions.
>
> In contrast, SANODEP takes a standard approach by constructing a carefully designed prior distribution to infer system properties during optimization. This choice is motivated by the specific requirements of our optimization problem. A key focus of SANODEP is the optimization of initial conditions (stage 1 optimization), without having any additional trajectory-level information. While maintaining trajectory-level information may potentially benefit the second-stage optimization in our BO framework, it does not directly contribute to the upstream problem  (initial condition optimization), as a result, if OOD happens on system dynamics, it is less beneficial to guard against trajectory level predictive performance as the first stage is already misleading. We believe an extensive investigation of robustness in such a setting is worth future investigation.
>
> > Sensitivity to the choice of hyperparameters
>
> SANODEP largely inherits NODEP's structure, minimizing the number of hyperparameters. All are set the same as NODEP, as detailed in Appendix 1. As for the loss function, we have demonstrated the optimization performance on two different scenario-mixing probabilities (i.e., $\lambda=0$ and $\lambda=0.5$) and demonstrate the performance difference is marginal in most benchmarking cases. We have also added a description in the paper about this aspect.
>
> [1] https://arxiv.org/abs/0912.3995
> [2] https://proceedings.neurips.cc/paper/2018/hash/41f860e3b7f548abc1f8b812059137bf-Abstract.html
> [3] https://arxiv.org/abs/1912.00967
> [4] https://proceedings.mlr.press/v202/fotiadis23a.html
>
> We thank the reviewer again for the valuable comments and have included the aforementioned explanations in the updated paper. We would like to clarify any aspects of unaddressed concerns upon further requests.

---

> > ### Comment · Reviewer_XihC · 2025-01-01
> >
> > The author addressed my concerns and I am inclined to accept.

---

### Review · Reviewer_2ir2 · 2025-03-04

**Summary Of Contributions:**

- This study addresses optimizing initial conditions and termination time in dynamical systems governed by unknown ODEs, with costly evaluations and delayed state measurements.
- Few-shot Bayesian optimization using prior information is introduced to efficiently identify optimal conditions in limited trials.
- System-aware neural ODE processes (SANODEP) is developed, expanding neural ODE processes to meta-learn ODE systems via a novel context embedding block.
- A two-stage Bayesian optimization framework is proposed to incorporate search space constraints, optimizing initial conditions and observation timings effectively.
- Experiments demonstrate SANODEP's potential in few-shot Bayesian optimization, while exploring its adaptability to varying levels of prior information.

**Audience:**

Yes

**Broader Impact Concerns:**

I don't think that this work has broader impact concerns.

**Claims And Evidence:**

No

**Requested Changes:**

Please see the textboxes above.

**Strengths And Weaknesses:**

Strengths

- It tackles an interesting application of Bayesian optimization to ODE systems.

Weaknesses

- Technical contributions are somewhat limited.
- Baseline methods are outdated.
- Intuition behind the components included in the proposed algorithm is missing.

Questions and Suggestions

- What is the meaning of a curly parenthesis in Equation (1)?
- Why is it a multi-objective optimization problem?  Why do you want to minimize evolution termination time?
- In Equation (2), maximize can be just $\max$.
- For the sentence "Simply put, we wish to maximize objective g early," please check if it is grammatically correct.
- Every equation is considered as a sentence, so that it should end with comma (,) or period (.). For example, Equations (1), (2), and (3) don't end with either , or ..
- Could you describe the difference between Equations (5) and (8) in plain text?  I believe that this should be added to improve the presentation.
- I think "section. 6.1" and "Appendix. A.3" should be "Section 6.1" and "Appendix A.3."  Specifically, there should be no period.  Also, Fig. is odd to me.  It can be just Figure.  There are too many similar issues; please fix all.
- In Section 3.1, why do the authors use DeepSets?  There exist more sophisticated set-taking methods like methods with different pooling methods.
- In Page 6, "can obtained" should be "can obtain."
- Basically, I don't understand Section 4.1.  Why do we need to find an initial condition using Equation (9)?  Isn't it given by practitioners?
- In Page 7, "Eq. 9,10" should be Equations (9) and (10).
- In Section 4.2, do the authors utilize the vanilla qEHVI?  Which implementation did you use?
- The title of Section 5, "Related Works" should be "Related Work."  Work is uncountable in this usage.
- Why do the authors use GP with the Matern 3/2 kernel?  Which implementation did you use for GP?
- In Page 9, "Table 2 and Table 3" should be "Tables 2 and 3."
- Font size of the caption of Table 1 is too small.
- I think NP and vanilla GP with the Matern 3/2 kernel are somewhat outdated.  The authors should try to compare more recent methods, for example, TNP and more sophisticated BO algorithms, to the proposed method.
- In Page 10, some paragraph heads have colon.  I believe they are not consistent.
- In Page 10, question marks for some paragraph heads are missing.
- In Page 10, "demonstrate" should be "demonstrates."
- I think that Algorithms 1 and 2 can be located in the main article, not in the appendix.  They are important to understand the proposed algorithm.

---

> ### Author Response · Authors · 2025-03-05
>
> We thank the reviewer for the thorough feedback and suggestions. We have updated the manuscript to incorporate these comments. Below is our point-by-point response:
> >  Why is it a multi-objective optimization problem? Why do you want to minimize evolution termination time?
>
> This problem has practical motivations in applications, for instance, in chemical reaction optimization, where the goal is to identify reaction configurations (initial conditions) that not only achieve high yield but also reach a satisfactory yield within a relatively short period of time (evolution termination time).
>
> >Could you describe the difference between Equations (5) and (8) in plain text?
>
> Equations (5) and (8) both represent the ELBO but differ in how the meta-training data is structured and how trajectory information is utilized. More specifically:
> - Equation (5): The model is trained on one trajectory at a time. NODEP learns to predict state values along an observed trajectory from a dynamical system, using only the context information from that specific trajectory. As a result, the model learns to interpolate or extrapolate states based on a limited set of observations from a single system realization.
> - Equation (8): In contrast, SANODEP generalizes NODEP by incorporating information from a batch of trajectories, each with different initial conditions but governed by the same dynamical system. While the goal remains to predict state values for a new trajectory, the model learns system-level properties from multiple trajectories. Additionally, the model can either interpolate new trajectories or perform pure forecasting when only the initial condition is provided. This is achieved using a Bernoulli variable in an amortized fashion.
>
> We have added a concise explanation in the updated manuscript.
>
> > In Section 3.1, why do the authors use DeepSets? There exist more sophisticated set-taking methods like methods with different pooling methods.
>
> We thank the reviewer for this insightful comment. We chose DeepSets for three key reasons:
> - It is computationally efficient, which we will further elaborate in our response below to the comparison with other methods.
> - This ensures that the performance improvement in NODEP arises from the incorporation of system information rather than from modifications to the pooling mechanism.
> - The feature extraction from multiple irregularly sampled time series remains a broader open challenge due to the difficulty of modeling temporal dependencies across irregular observations, and set-based representations in general may not necessarily be the most principled approach in this setting.
>
> > Basically, I don't understand Section 4.1. Why do we need to find an initial condition using Equation (9)? Isn't it given by practitioners?
>
> Finding an optimal initial condition using Equation (9) is necessary because the initial condition directly influences the trajectory of the dynamical system, ultimately determining the final state and the objective function $g(\cdot) $. While practitioners may specify an initial condition, it is not necessarily optimal. Therefore, the goal of Section 4.1 is to actively search for the best initial condition that maximizes $g(\cdot) $ rather than relying on a predefined one.
>
> > In Section 4.2, do the authors utilize the vanilla qEHVI? Which implementation did you use?
>
> We use the vanilla qEHVI implementation, which calculates the hypervolume via the inclusion-exclusion principle [1]. We use the implementation from  `Trieste`.
>
> > Why do the authors use GP with the Matérn 3/2 kernel? Which implementation did you use for GP?
>
> GP is of standard choice in BO and the Matérn 3/2 kernel is one of the standard kernel choices for GPs. As mentioned in the updated manuscript, we use the implementation from `Trieste`.
>
> -------------
> [1] https://arxiv.org/abs/2006.05078

---

> > ### Author Response · Authors · 2025-03-05
> >
> > > I think NP and vanilla GP with the Matérn 3/2 kernel are somewhat outdated. The authors should try to compare more recent methods, for example, TNP and more sophisticated BO algorithms, to the proposed method.
> >
> > We appreciate this suggestion. The primary focus of our paper is to develop an ODE-based model for few-shot Bayesian optimization (BO) in dynamical systems. Our experimental scope is restricted to GPs, NPs, and NODEP for the following reasons:
> > - **TNP is not inherently designed for modeling dynamical systems**
> > While Neural Process (NP) models like TNP have demonstrated strong empirical performance in emulating GPs and image-based tasks, they are not inherently suited for modeling and optimizing dynamical systems. The differences in our optimization setting include:
> >     - **Context Set Evolves Dynamically** Unlike standard NP prediction tasks, where the context set $C$ remains static, in our setting, the context set is dynamically updated with new trajectories as different initial conditions are known.  This requires recomputation of self-attention representations at every step, making complex context encoding (like self-attention in TNP) computationally expensive.
> >
> >     - **Scalability Limitations of TNP for Large Context Sets**: Our training model handles a dataset of $\sum_{j=1}^{\boldsymbol{N}_{\boldsymbol{x}_0}}| \mathcal{T}_j |$, where data is subsampled into context and target sets dynamically with masking.
> >         - Modern deep learning frameworks (e.g., `JAX`) do not yet efficiently support sparse attention masks to reduce computation complexity.
> >       - Computing self-attention over such large context sets (with trajectories from multiple initial conditions, $\sum_{j=1}^{\boldsymbol{N}_{\boldsymbol{x}_0}} | \mathcal{T}_j |$ in total) is computationally prohibitive.
> >
> > - **Existing BO Algorithms Do Not Address Our Problem Setting**
> > As elaborated in Section 4, our optimization problem presents novel challenges that standard BO algorithms do not naturally address. These challenges make a direct comparison impractical and motivate the development of Algorithm 2, which is explicitly designed to handle evolving trajectory-based search spaces.
> > - **Why We Focus on ODE-Based Models**
> > Our primary claim is that incorporating system dynamics explicitly enhances few-shot BO in dynamical systems, making ODE-based models a more natural fit for our setting.
> >     - ODE-based models provide additional functionality, such as modifying trajectory dynamics, which is essential for optimizing dynamical systems (Appendix E.1). This flexibility allows for direct integration with the underlying physics, unlike purely data-driven models such as TNP.
> >     - An end-to-end comparison with TNP would be interesting, but as elaborated, the integration of TNP into our optimization framework would require substantial modifications to handle evolving context sets and high-dimensional trajectory data, which we leave for future work.
> >
> > > - I think "section. 6.1" and "Appendix. A.3" should be "Section 6.1" and "Appendix A.3." Specifically, there should be no period. Also, Fig. is odd to me. It can be just Figure. There are too many similar issues; please fix all.
> > > - In Page 6, "can obtained" should be "can obtain."
> > > - In Page 7, "Eq. 9,10" should be Equations (9) and (10).
> > > - The title of Section 5, "Related Works" should be "Related Work." Work is uncountable in this usage.
> > > - In Page 9, "Table 2 and Table 3" should be "Tables 2 and 3."
> > Font size of the caption of Table 1 is too small.
> > > - In Page 10, some paragraph heads have colon. I believe they are not consistent.
> > > - In Page 10, question marks for some paragraph heads are missing.
> > > - In Page 10, "demonstrate" should be "demonstrates."
> > > - In Equation (2), maximize can be just .
> > > - For the sentence "Simply put, we wish to maximize objective g early," please check if it is grammatically correct...
> >
> > We appreciate the reviewer’s detailed feedback on formatting consistency. We have carefully reviewed and corrected all identified issues in the updated manuscript.
> >
> > > - What is the meaning of a curly parenthesis in Equation (1)?
> >
> > We are trying to make curly parenthesis to denote the initial condition is specified by $\boldsymbol{x}_0$ for this ODE, but we have changed to a more clear notation in Equation (1).
> >
> > > I think that Algorithms 1 and 2 can be located in the main article, not in the appendix. They are important to understand the proposed algorithm.
> >
> > We appreciate this suggestion. Given the importance of Algorithm 2 for understanding our optimization framework, we have moved it to the main paper to enhance clarity. However, due to space constraints, Algorithm 1 remains in the appendix while ensuring all necessary details are referenced in the main text.
> >
> > -------------
> >
> > We hope the above response and updated manuscript address the reviewer’s concerns and are happy to clarify any remaining aspects upon request.

---

### Decision · Action_Editor_TbAG · 2025-04-03

**Recommendation:** Accept as is

**Comment:**

This submission considers the problem of optimizing initial conditions and termination time in dynamical systems governed by unknown ordinary differential equations (ODEs). It takes the ODE approach, extends the NODEP method, proposes a few-shot Bayesian optimization  framework to incorporate system's prior information.

Some reviewers consider it to address a relevant and practical challenge in optimizing dynamical systems with limited evaluative resources. The extension of NODEP to the meta-learning setting with system awareness improves learning efficiency and shows some level of novelty. However, reviewers consider the technical contribution to be limited and the scope of the problem is relatively narrow. The empirical comparison to the limited set of GP and NP-based meta-learning methods remains a concern to reviewers even though the authors explain why other more recent models and standalone approaches do not apply or require a significant effort to be adapted to the problem under study.

The authors provide a revision to address formatting issues and provide more clarification following the reviewer discussion. While some reviewers are still concerned of its limited scope and baselines, they all agree that the proposed method has demonstrated its value in addressing the specific optimization problem of dynamic systems.

**Audience:**

This submission would be of interest to a limited set of audience working on dynamic system modelling, optimization, and meta-learning approaches.

**Claims And Evidence:**

This submission provides thorough experiments in a variety of dynamical systems to demonstrate the potential of the proposed SANODEP method in few-shot Bayesian optimization for dynamic systems, compared to baselines of GPs, NPs, and NODEP. It provides implementation details of the baselines, and discussion how the set of baselines are chosen and why other candidates (e.g. TNP, GP-ODE, and other non-meta learning methods do not apply to the problem under study).